# Dynamical Equations With Bottom-up Self-Organizing Properties Learn Accurate Dynamical Hierarchies Without Any Loss Function

## Abstract

Self-organization is ubiquitous in nature and mind. However, machine learning and theories of cognition still barely touch the subject. The hurdle is that general patterns are difficult to define in terms of dynamical equations and designing a system that could learn by reordering itself is still to be seen. Here, we propose a learning system, where patterns are defined within the realm of nonlinear dynamics with positive and negative feedback loops, allowing attractor-repeller pairs to emerge for each pattern observed. Experiments reveal that such a system can map temporal to spatial correlation, enabling hierarchical structures to be learned from sequential data. The results are accurate enough to surpass state-of-the-art unsupervised learning algorithms in seven out of eight experiments as well as two real-world problems. Interestingly, the dynamic nature of the system makes it inherently adaptive, giving rise to phenomena similar to phase transitions in chemistry/thermodynamics when the input structure changes. Thus, the work here sheds light on how self-organization can allow for pattern recognition and hints at how intelligent behavior might emerge from simple dynamic equations without any objective/loss function.

## 1 Introduction

Self-organization is present in diverse scientific fields, from biology (Misteli, 2007; Deglincerti et al., 2016; Sasai, 2013) to neuroscience (Linsker, 1988; Tognoli & Kelso, 2014; Imam & L. Finlay, 2020; Schoner & Kelso, 1988), chemistry (Montalti et al., 2017; Lehn, 2002a;b) and physics (Haken, 1975; Wickman & Korley, 1998; Tersoff et al., 1996; Haken, 1977). It shows how order can arise intrinsically from a system. It is a set of interactions that allows for the emergence of patterns and is responsible for complex behavior from simple interactions (Kauffman et al., 1993; Haken, 1977). Albeit the ubiquitous presence of self-organization in nature and in the brain, it is unknown how self-organization can lead to intelligence. For this reason, theories of intelligence rarely use the concept in their development. The free energy principle (Friston, 2010; 2009) and reinforcement learning paradigms (Sutton & Barto, 2018; Mnih et al., 2015; Schrittwieser et al., 2020) define a top-down view of learning based on objectives that are satisfied locally or globally. However, from a bottom-up perspective, it is still barely understood how Hebbian learning (Hebb, 2005; Magee & Johnston, 1997) and other neuron behaviors allow for top-down theories of intelligence to emerge. In fact, there is strong evidence the brain does not behave as a computer but as a more self-organizing system (GRAY, 1987; Eckhorn et al., 1988). In this paper, we show how the learning of patterns can be achieved by Hebbian and anti-Hebbian learning dynamics, linking between Hebbian learning and top-down theories of intelligence (Hebb, 2005).

The recent success of machine learning, similar to the current theories of intelligence, is mostly given to optimization-based deep learning algorithms. While deep learning utilizes optimization and loss functions (objective functions) to learn the model's parameters and improve in the task at hand, self-organization existence in machine learning is mostly limited to Self-Organizing Map (SOM) variations (Kohonen, 1982; Chang et al., 2020; Reker et al., 2014). Such SOMs are only

employed in clustering and dimensional reduction tasks, as they lack the ability to find patterns in data required for further processing and acting on the environment.

Here, inspired by many successful modelings of neuron behaviors based on dynamical equations composed of attractor dynamics (Tognoli & Kelso, 2014; Wills et al., 2005; Spalla et al., 2021; Ooi et al., 2018), we show how a system of dynamical equations can give rise to order and represent patterns. Our proposed system is arguably more biologically plausible, and it is also shown to be more accurate and adaptive than state-of-the-art unsupervised algorithms. In fact, it sets up a foundation for a new paradigm in machine learning solely based on self-organization from dynamical equations, namely Self-Organizing Dynamical Equations, which are inherently accurate and adaptive. We propose Hierarchical Temporal Spatial Feature Map (TSFMap), a learning system implementing the Self-Organizing Dynamical Equations paradigm. It creates a space in which distances in it reflect the temporal correlation between input variables. A simple clustering in this self-organized space reveals that the representation learned is very accurate. Adaptation comes from the fact that the proposed system, Hierarchical TSFMap, couples its internal dynamics with the input, resulting in patterns encoded as emergent attractor-repellers at equilibrium. Consequently, alterations in the underlying structure of the problem result in different equilibrium with new attractor-repellers, triggering an inherent adaptation when the problem changes. Interestingly, structural changes in the environment cause in Hierarchical TSFMap a phenomenon very similar to phase transition observed in thermodynamics, and chemistry, among other areas (Fig. 1).

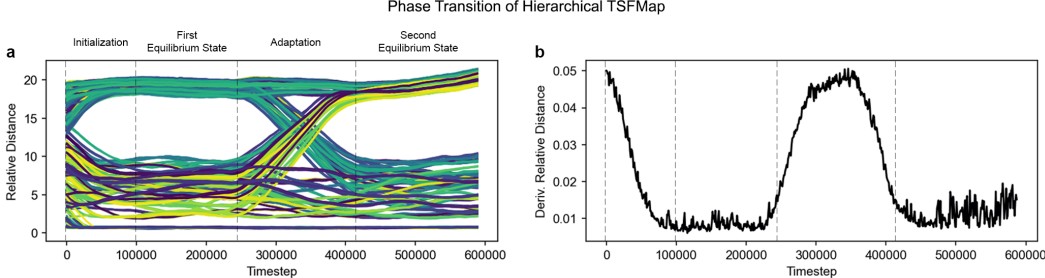

Figure 1: Hierarchical TSFMap's phase transition. A phenomenon similar to phase transition takes place in the proposed algorithm when the underlying structure of the problem changes. (a) Lines indicate the relative distance for all weight pairs. (b) The average rate of change for all weight pairs' distances (a). Random initialized weights start to form patterns with respect to the input and enter an equilibrium state. Once the problem's data structure is altered, Hierarchical TSFMap automatically adapts its weights. Subsequently, weights enter another equilibrium state.

In this paper, Hierarchical TSFMap is evaluated in one of the hardest types of patterns, e.g., recognition of dynamical and imbalanced hierarchical patterns present in sequential data. The problem of learning the hierarchical relationships from sequential input is a challenging unsolved one (Uddén et al., 2020). This becomes even harder when the problem structure is dynamic, e.g., variable correlations change over time. Since any information can be serialized, the pattern recognition over sequences is a general one that can be applied ubiquitously to any type of serialized data. Albeit the difficulty of the task, Hierarchical TSFMap provides, perhaps surprisingly, near-optimal solutions to more than half of the problems. Lastly, we have demonstrated that Hierarchical TSFMap can extract hierarchical structures from sequential data generated from two real-world networks: (1) Zachary's karate club network and (2) Lusseau's bottlenose dolphin social network.

## 2   RELATED WORK

A recent work (Vasconcellos Vargas & Asabuki, 2021) demonstrated how a self-organizing system called SyncMap, can learn features from sequences using dynamical equations alone (e.g., without any type of optimization). Here we go beyond this work on simple chunks to show how dynamical equations that self-organize compose a paradigm and can be used to deal with challenging hierarchical structures and imbalanced problems. In fact, the experiments suggest that Hierarchical TSFMap can deal with dynamical variations of the problems with little difficulty.

Community detection in complex networks can also extract hierarchies (Clauset et al., 2007; Corominas-Murtra et al., 2013). Although the input data, and therefore the problem, is different from the one seen here, sequence data and complex networks can interchangeably convert to one another (e.g., via an adjacency matrix from transition probabilities or a random-walk over a complex network). This reveals Hierarchical TSFMap's connection with complex networks. Having said that, the similarities stop here as both the objective and methodology differ. Complex networks' algorithms usually maximize a metric on the network to find communities (while here only dynamical equations are used). Given the nature of optimization problems, such models are inherently not able to deal with any possible dynamics of the network.

A closely related body of work is that of learning an embedding that also preserves the variables' correlations. Word2vec, specifically, can create embeddings that preserve the relationships of neighboring variables based on their context (Mikolov et al., 2013). However, we showed here that hierarchical structure does not seems to be preserved in this embedding. To make matters worse, adaptation is tricky with deep neural networks (it is in direct conflict with techniques that make them learn well such as decreasing learning rate) and there is no inherent system that can adapt to changes in the environment.

## 3 Hierarchical Temporal Spatial Feature Map

Here we demonstrated the general workflow of Hierarchical TSFMap, which is composed of three steps: (1) input encoding, (2) the dynamics process, and (3) the hierarchical chunking phase. Refers to Fig. 2 for an overview of Hierarchical TSFMap's workflow.

**Input Encoding.** We first encoded the sequence data generated from the problem into a specific type of input before feeding it into our model. Given a sequence of data $X$ with $\tau$ be the sequence length, all unique items in $X$ represent different states and we denote the total number of unique states using $n$. We converted input sequence into a sequence of state $S_t = (S_1, S_2, \cdots, S_\tau)$. $S_t$ be a vector of state values in time $t$. We set $s_t = s_{1,t}, s_{2,t}, \cdots, s_{n,t}$ with $s_t \in S_t$ and total number of unique state $n$ as the dimension of states. For $s_t \in \{0,1\}^n : \sum_{i=1}^{n} s_{i,t} = 1$, simulating the activation of neurons. The input encoding is modeled as an exponentially decaying vector $x_t$, sharing the same size as the number of states:

$$x_{i,t} = \begin{cases} s_{i,ta} \times e^{-0.1 \times (t-ta)}, & t - ta < m \times tstep \\ 0, & otherwise \end{cases} \tag{1}$$

in which $ta$ is the most recent state transition to state $s_i$. State transitions happen every $tstep$ step and variables with time of activation greater than $m \times tstep$ are set to 0. Thus, only the last $m$ states activated are remembered as $x_{i,t} > 0$, and we set $m$ to 10. At each time step, $x_{i,t}$ will be fed into the model as a spike encoded input.

**Dynamics.** Hierarchical TSFMap represents patterns with a formation of attractor-repeller pairs for each identified one. The space made of attractor-repeller pairs defines a temporal to spatial mapping of variables' correlation, and is given the name $\sigma$ space. There is no optimization or objective function, the dynamical system merely self-organizes to the input, following positive and negative feedback loops. Experiments suggest that the distance between patterns in the learned $\sigma$ space is proportional to the strength of their temporal correlation.

To begin the dynamic process, all inputs $x_{i,t}$ have a set of corresponding weights $w_{i,t}$ initialized to a random position in a $\sigma$ space $w_{i,t} \in \mathbb{R}^k$ at the beginning, with $k$ be a hyperparameter that defines the dimension of the map, or the degrees of freedom that organize the weights.

Hierarchical TSFMap defines positive and negative feedback loops related to which state variables activate together (synchronous behavior). In each iteration $t$, state variables that activate or deactivate together are first included into $PS$ or $NS$ sets respectively. Here, $PS$ or $NS$ refers to: (1) activated and recently activated input set $PS_t$ and (2) non-recently activated input set $NS_t$. Inputs with value greater than or equal to 0.1 are considered an element of $PS_t$; otherwise, inputs are a member of $NS_t$. Thus, we define $PS_t = \{i | x_{i,t} > 0.1\}$ and $NS_t = \{i | x_{i,t} \leq 0.1\}$. If and only if the cardinality of both sets are greater than one, where $|PS_t| > 1$ and $|NS_t| > 1$, the centroid of both sets are computed as follows (otherwise no update is made in this iteration):

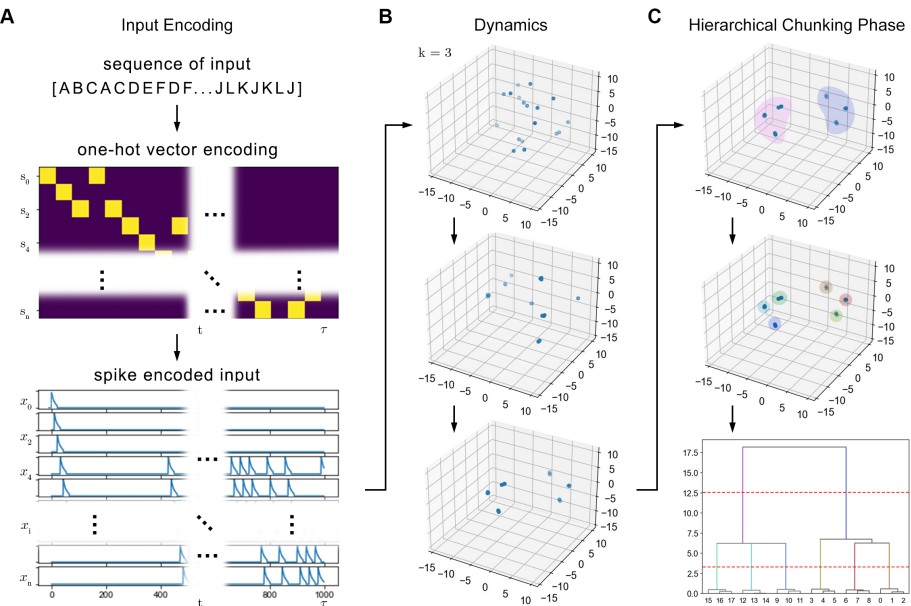

Figure 2: Hierarchical TSFMap's workflow. (a) A sequence of variables is converted to spikes that decay exponentially. (b) Hierarchical TSFMap's weights are initialized randomly with a weight for each possible variable. Every time step the spike encoded input is presented to the algorithm which self-organizes to it. (c) The $\sigma$ space stores the temporal relationship of variables spatially. To extract this hierarchical information into dendrograms, a simple hierarchical clustering is used.

$$cp_t = \frac{\sum_{i \in PS_t} w_{i,t}}{|PS|}, \quad cn_t = \frac{\sum_{i \in NS_t} w_{i,t}}{|NS|} \tag{2}$$

where $cp_t$ and $cn_t$ are the centroids of $PS_t$ and $NS_t$ respectively. With $cp_t$ and $cn_t$, we determine the distance of all weights to $cp_t$ and $cn_t$ as $d_{cp} = \|w_{i,t} - cp_t\|$ and $d_{cn} = \|w_{i,t} - cn_t\|$ respectively, using Euclidean distance metric. Subsequently, state variables are updated (Fig. 3) by either attracting to $cp_t$ (activated states) or repelling from $cn_t$ (inactive states).

$$v_{i,t+1} = \theta v_{i,t} + \left[1_{PS_t}(i)\frac{\mu_1(cp_t - w_i)}{d_{cp}} + 1_{NS_t}(i)\left(\frac{\mu_2(w_i - cn_t)}{d_{cn}} + \frac{\mu_3(w_i - cp_t)}{d_{cp}^2}\right)\right] \tag{3}$$

$$w_{i,t+1} = w_{i,t} + \alpha v_{i,t+1} \tag{4}$$

where $\alpha = 1e - 3$ is the learning rate, $\theta = 0.999$ is the velocity decay and $v$ is the velocity. $1_{PS_t}(i)$ (or $1_{NS_t}(i)$) is the indicator function that maps elements of the subset $PS_t$ (or $PN_t$) to one, and all other elements to zero. The term $1_{PS_t}(i)\frac{\mu_1(cp_t - w_i)}{d_{cp}}$ acts as an attraction force between activated variables; On the other hand, $1_{NS_t}(i)\left(\frac{\mu_2(cn_t - w_i)}{d_{cn}} + \frac{\mu_3(w_i - cp_t)}{d_{cp}^2}\right)$ acts on in-active variables as an attraction force between in-active variables and repulsion force from activated variables. $\mu_1 = 6$, $\mu_2 = 3$, and $\mu_3 = 2$ are the coefficients that control the strength of the attraction and repulsion forces, tuned from a range of values that were tried as they produced the best results. This dynamic law governed by the Hebbian and anti-Hebbian learning dynamics is arguably analogous to a force-directed algorithm (Fruchterman & Reingold, 1991), or even a gravity and anti-gravity force. We then specify the attraction force ($F1$) with $1_{PS_t}(i)$ indicating that $F1$ only appears among the activated state variables. For the inactive ones ($1_{NS_t}(i)$), we specify the attraction force between inactive variables ($F2$) and a repulsion force from activated variables ($F3$). Weights (e.g., state variables) are finally updated by Eq. 4. Each update iteration ends by scaling all weights to a fixed size space. Furthermore, the velocity parameters create inertia to avoid the instability caused

by instantaneous weight update. At the end of the iteration, all values of the updated weights are normalized: $\hat{w}_{i,t+1} = \frac{w_{i,t+1}}{max(w)}$ to keep them in a relative space. Overall, the dynamical equation was found to work well in preserving the hierarchical structure of the corresponding input, despite its simplicity. The process above iterates until the final time step. Theoretically, a final time step is not required to be defined, as this is an adaptive system.

$$v_{i,t+1} = \theta v_{i,t} + \Big[ 1_{PS_t}(i) \underbrace{\frac{\mu_1(cp_t - w_i)}{d_{cp}}}_{F1} + 1_{NS_t}(i) \Big( \underbrace{\frac{\mu_2(w_i - cn_t)}{d_{cn}}}_{F2} + \underbrace{\frac{\mu_3(w_i - cp_t)}{d_{cp}^2}}_{F3} \Big) \Big]$$

$$w_{i,t+1} = w_{i,t} + \alpha v_{i,t+1}$$

Figure 3: Hierarchical TSFMap's main dynamical equations and emerging behavior. The circles with arrows represent the emergent attractor-repeller pairs (or just repellers); a consequence of the dynamical equations. Regarding the equations, $v_{i,t}$ is the velocity with $\theta$ as its decaying factor at time $t$, while $w_{i,t}$ is the weight. The positive centroid ($cp_t = \sum_{i \in PS_t} w_{i,t}/|PS|$) attracts recently activated weights ($F1$) and repels inactive weights ($F3$); while the negative centroid $cn_t = \sum_{i \in NS_t} w_{i,t}/|NS|$ only repels inactive weights ($F2$). Distance of all weights to $cp_t$ and $cn_t$ are computed as $d_{cp} = \|w_{i,t} - cp_t\|$ and $d_{cn} = \|w_{i,t} - cn_t\|$ respectively using Euclidean distance metric. With $\alpha$ as learning rate, $w$ is subsequently updated with regard to $v$.

**Hierarchical Chunking Phase.** Methods like hierarchical clustering can produce a dendrogram. Yet, a dendrogram does not promptly reveal which level of the hierarchy should be viewed as a collection of meaningful chunks. To solve that, we use Hierarchical Chunking Phase to extract the information of how variables are chunked together on each level of the hierarchy. With Hierarchical Chunking Phase, the proposed algorithm produces an $L \times N$ matrix, which refers to the output $Y$, including the predicted class label, with $L$ being the total levels of hierarchy (see Appendix. A for the implementation detail of Hierarchical Chunking Phase).

## 4 EXPERIMENTS AND RESULTS

We investigate the performances of Hierarchical TSFMap and the baselines (SyncMap, Word2vec, Modularity Maximization, and transition probability matrix) in two types of hierarchical problems: imbalanced hierarchies and dynamical hierarchies (hierarchies that change during experiments). Each problem is represented by a graph preserving the hierarchical structures. Each sequence observed by the algorithms is derived from a random walk in the above-mentioned graph, with decreasing transition probabilities when variables pertain to different chunks. Variable is placed into the sequence input whenever random-walker travel to one. Refer to Appendix. B for the implementation of the baselines and Appendix. C for the details of graph-to-input-sequence generation.

### 4.1 IMBALANCED HIERARCHICAL STRUCTURE

Real-world events rarely share equal possibilities, suggesting that most of the real-world structures are arguably imbalanced. We first introduce three environments to quantify the performances of the models on imbalanced data: Imbalanced Hierarchy (IH), Hierarchy with Branches (HB), and Imbalanced with Extra Hierarchy (IEH). These are generated from three graphs indicating the desired imbalanced structures (Fig. 4). The environments create a distribution of variables where the occurrence of some variables is more frequent than others. In detail, IH defines a sequence where a single chunk contains much more variables than other chunks, while HB is a hierarchical structure where a branch has a shallower hierarchy with fewer nodes. With more complexity, IEH has a branch with deeper hierarchical structure.

Results shown in Fig. 4 reveal that Hierarchical TSFMap surpasses all other algorithms in imbalanced hierarchical problems. This suggests that self-organization alone can, perhaps surprisingly, represent such complex structures. Hierarchical TSFMap's behavior also resembles the bottom-up behavior observed in natural self-organization processes (Simon, 1991) (Fig. 5). Weights tend to

form chunks that belong to the lower level of hierarchies at first. These chunks then proceed to cluster into bigger chunks that belong to the next level of the hierarchy. It is important to note that weights manoeuvre and form chunks around the surface of a $k-1$ dimensional n-sphere. Such an n-sphere composition allows for negative centroids ($cn_t$) mostly at the center of the n-sphere and for positive centroids ($cp_t$), when correctly clustered, to be at the border. The result is a uniform negative feedback ($F2$) away from the center and a non-uniform positive feedback perpendicular to the center ($F1$). Notice that, since all weights are scaled back to a fixed size space, the negative feedback $F2$ is canceled, bringing the system to equilibrium (only $F1$ and $F3$ move weights respectively close and far apart from each other on the border of the n-sphere; proportionally to their temporal correlation). Important to notice that $F2$ allows for a degree of freedom (e.g., they are not fixed at the border of the n-sphere) for weights to move around while keeping them mostly stable in equilibrium. See Fig. 13 for the visualization of Hierarchical TSFMap's dynamic in Imbalanced Hierarchical Structure problems.

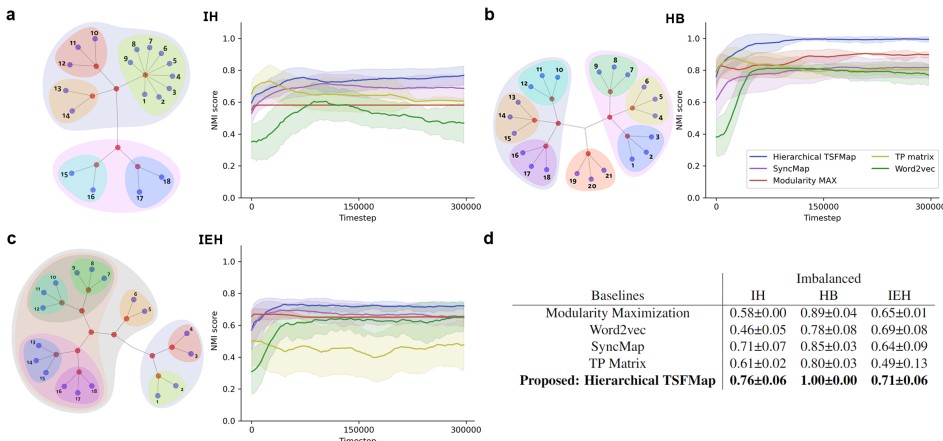

Figure 4: Experiment setting and results of imbalanced hierarchical structure (HB). (a, b, c) The graphs of hierarchical structures (left) of the environments IH, HB and IEH used to generate input sequences. Leaf nodes represent variables, and sibling nodes with the same color coding belong to the same chunk linked by the nodes in red. A chunk can contain either child chunks or variables. The plot (right) illustrated the Normalized Mutual Information (NMI) score progression over time. The solid lines and shade represent the mean and standard deviation of the NMI score over 30 instances. Results are smoothed by a ten-timestep moving average. (d) The NMI score of Hierarchical TSFMap compared to baselines in table view. The result indicates that Hierarchical TSFMap performs the best in all imbalanced hierarchical structure experiments.

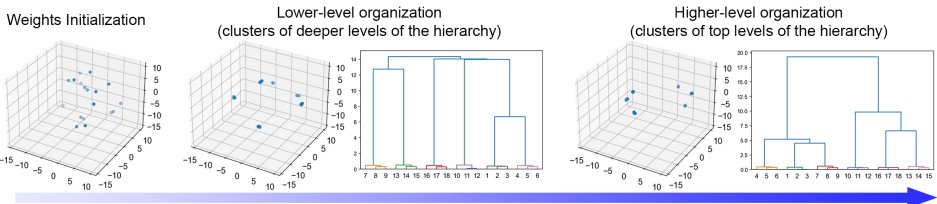

Figure 5: Analyzing Hierarchical TSFMap's Dynamics. Weights are initialized randomly in a three-dimensional space ($k = 3$). The weights progressively self-organize into six smaller chunks, these smaller chunks proceed to then merge into two bigger chunks respectively. This emerging behavior reveals the bottom-up self-organization properties of Hierarchical TSFMap, in which individual components gradually aggregate to form more complex systems iteratively.

To understand the reason for the accurate results from Hierarchical TSFMap when compared to other embedding-based learning systems such as Word2vec and SyncMap, we compared their learned embeddings/maps in Fig. 6. Specifically, the $\sigma$ space learned by Hierarchical TSFMap shows it can

learn the temporal correlation between variables. In fact, the learned $\sigma$ space respects both local and global temporal correlations. Word2vec is shown able to identify local chunks with substantial accuracy, but global relationships are not preserved in its embedding. The rationale behind this lies in how Word2vec learns, e.g., by using local contextual information which is less predictive of global contexts/relationships. SyncMap, on the contrary, can identify high-level chunks precisely. However, there seems to be a scaling problem in how local chunks are clustered, that is, local chunks tend to overlap with each other, making it difficult to accurately identify the lower level structure of a given hierarchy. Regarding the TP matrix, the precision of the transition probability's table is affected strongly by the standard deviation of variables. This problem further increases in cases with smaller chunks that have a smaller probability of activating, justifying the poor performance. This is also the case of Modularity Maximization which is also based on the transition probabilities. Moreover, researchers have already shown that the used *modularity* metric tends to overestimate either the global context or local context of a chunk (Sun, 2016).

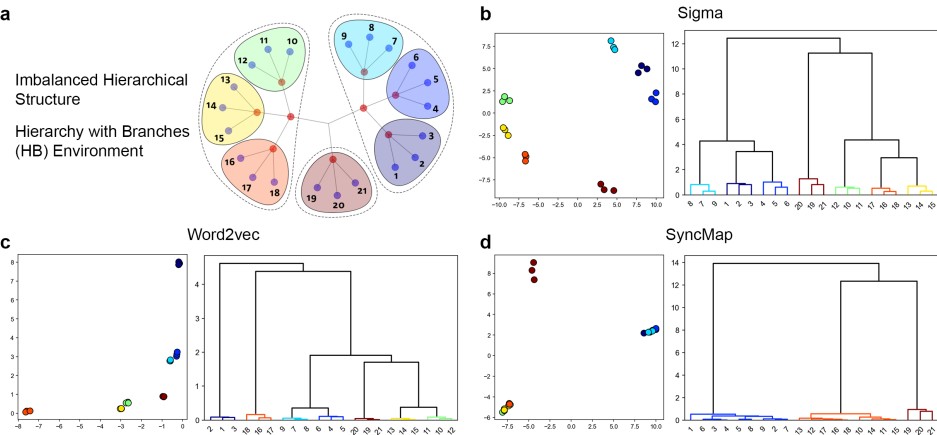

Figure 6: Comparison of the Learned Representation of Hierarchical TSFMap, SyncMap, and Word2vec. Here we show (b) the 2D learned $\sigma$ space of Hierarchical TSFMap, (c) the learned map of SyncMap, and (d) the word embedding of Word2vec; together with the dendrogram formed corresponding to their pattern in (a) HB environment. Chunks in the lowest level of hierarchies are color-coded. As shown in (b), Hierarchical TSFMap can produce a pattern that matched the distribution of variables. (c) The word embedding learned by Word2vec can identify local chunks, yet failed to identify chunks beyond the lowest level of hierarchies. (d) SyncMap successfully identifies chunks on a global scale; however, local chunks overlapped, which increases the difficulty to distinguish them.

## 4.2 DYNAMIC HIERARCHICAL STRUCTURE

Real-world problems are constantly changing. Yet, humans adapt to it almost effortlessly while understanding complex hierarchical relationships (Conway & Christiansen, 2001; Werchan et al., 2015; Collins & Frank, 2013). To quantify the performance of algorithms under problems with hierarchies that change over time (dynamical hierarchies), five problems with different characteristics are defined (Fig. 7). In detail, DIH (Dynamic Imbalanced Hierarchy) starts with HB's imbalanced hierarchical structure and then merges two chunks into a new branch. DCH (Dynamic Chunk Hierarchy) splits two chunks into three chunks over time. EC2EH (Extra Chunk to Extra Hierarchy) shifts from shallow hierarchical structure with two levels to a deep hierarchical structure with four levels. EH2EC (Extra Hierarchy to Extra Chunk) is the reverse of EC2EH, specifically designed to test what happens when hierarchical structures decrease in the number of levels. DCS (Dynamic Chunk Swap) swaps chunks of level two to form a different structure. The distribution of variables in dynamic environments shifts over time (Fig. 9), halfway through the input sequence, when $\tau/2$ with a total number of input $\tau = 600000$ (This scheme applies to all the following dynamic problems). Note that the number of variables remains consistent despite the changes. The dynamic problems aim to evaluate how models can adapt to the latest changes in the environment.

Results show that Hierarchical TSFMap can adapt well in dynamic environments, achieving near-optimum solutions in 4 out of 5 environments. The experiments here extend the results with imbalance hierarchies to demonstrate that the good performance is not only limited to static problems. Moreover, the rapid remapping of the weights when an instantaneous change occurred in environments is analogous to the attractor dynamics of place cells, as they switch between representations to respond to the changes in environments (Wills et al., 2005).

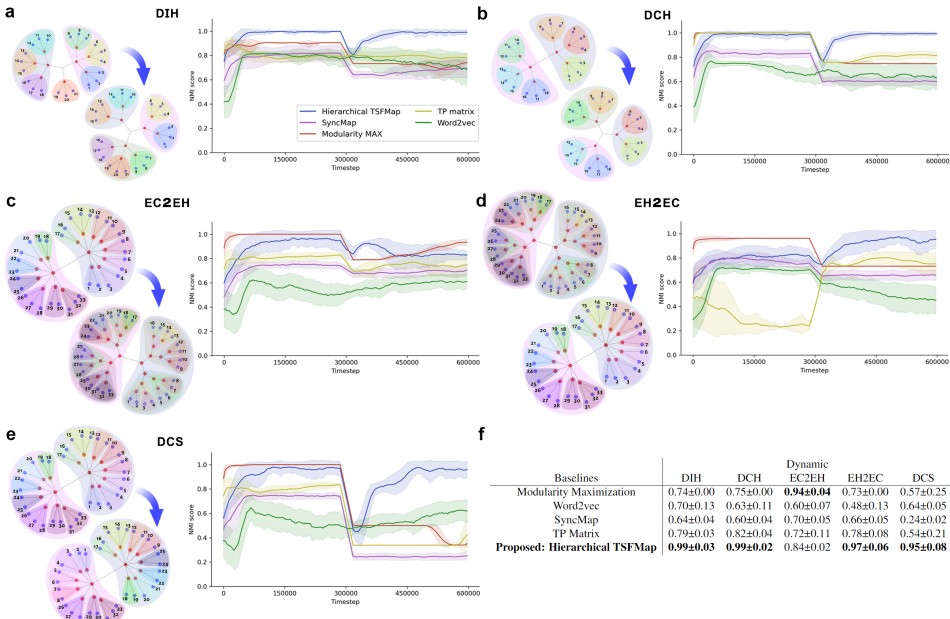

Figure 7: Experiment setting and results of dynamic hierarchical structure. (a, b, c, d, e) The graphs of hierarchical structures (left) of the environments DIH, DCH, EC2EH, EH2EC, and DCS used to generate input sequences. The blue arrow implies that the distribution of variables changes throughout a single experiment. The plot (right) illustrated the NMI score progression over time. The proposed method, Hierarchical TSFMap, is shown to be the only one capable of quickly adapting to structural changes. (f) The table showed the NMI score of Hierarchical TSFMap compared to baselines. The proposed algorithm performs the best in nearly all dynamic hierarchical structure experiments.

In fact, when compared with other methods, Hierarchical TSFMap shows a great performance before and after the change in structure (Fig. 7). Much of the great performance derives from phase transitions that happen naturally in Hierarchical TSFMap when the input structure changes (Fig. 1). See Fig. 14 and Fig. 15 for the visualization of Hierarchical TSFMap's dynamic in Dynamic Hierarchical Structure problems. All the other methods face different but related problems related to adaptation. TP matrix and Modularity Maximization are based on transition probabilities which become imprecise when the underlying probabilities change throughout the test. Word2vec has learned weights that become, after the change, a local minimum which is hard to overcome and bias the learning toward a previously learned nearby region. SyncMap was not designed for hierarchies (reflected by the relatively poor performance even in static problems). Increasing the difficulty of hierarchical problems with dynamical structural changes only makes matters worse. Additionally, although initialized in higher dimensional weight space, the rank of SyncMap's weight matrix converged to 1 given enough time, where $\rho(W) = 1$ with $W$ being the weight matrix. This indicates that SyncMap's dynamic can be restricted in one-dimensional space. The weight matrix of Hierarchical TSFMap however, can retain its high dimensionality, where $1 \leq \rho(W) \leq k$ (Fig. 12).

## 4.3 REAL WORLD SCENARIOS

In this section, we consider two network datasets with interpretable hierarchical structures: (1) Zachary's karate club network and (2) Lusseau's bottlenose dolphin social network. Despite being

well-establish benchmarks, the hierarchical information of both networks is seldom explored in depth. Therefore, we investigate the hierarchical structure extracted from Hierarchical TSFMap, utilizing the input sequence generated from the networks (refers to Appendix. D for the experiment details).

The ground truths provided by Girvan & Newman (2002); Zachary (1977) and Lusseau et al. (2003) are only available for one level of the hierarchy; Thus, the interpretation of the remaining hierarchical structure relies on the visualization of the representation space. The table in Fig. 8 showed the NMI score of the most significant chunk predicted by the models compared to the ground truth in both tasks. The result showed that our models could configure their weights to match the ground truth in most instances, reflected by the relatively high NMI score. Furthermore, Fig. 8 (c) demonstrated the weight space of Hierarchical TSFMap for the Karate club network. Two chunks are formed on the most significant level of the hierarchy, aligning with the ground truth where the members of the Karate club were eventually split into two factions. When looking deeper into the hierarchy, smaller groups of members and less social individuals are formed into their own chunks, showing signs of hierarchical structure in the network. Lastly, Fig. 16 and Fig. 17 displayed the representation space of Hierarchical TSFMap, SyncMap, and Word2vec in both tasks.

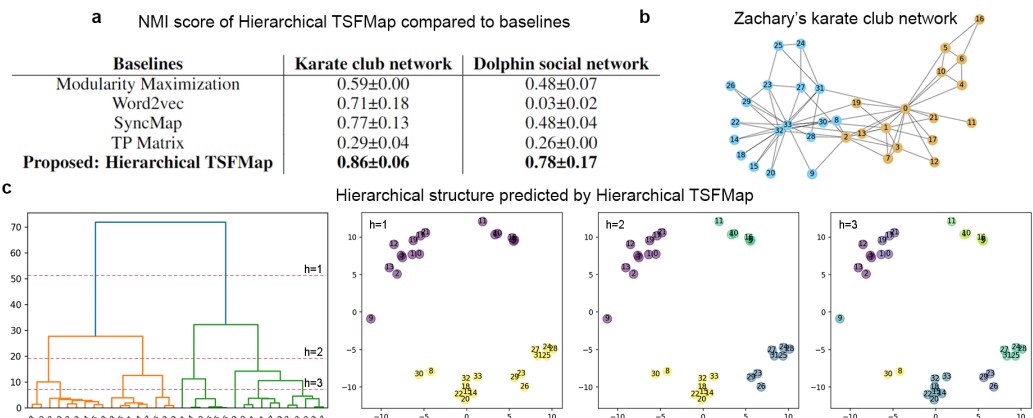

Figure 8: Hierarchical TSFMap can extract the hierarchical structure of real-world networks. (a) The NMI score of Hierarchical TSFMap compare to baselines on Karate club and Dolphin social network. (b) The ground truth of the Karate club network provided by Zachary (1977). (c) The dendrogram shows the hierarchical clustering of weights, where the red dashed lines are the cut-off. The weight space ($k = 5$) describes how weights are chunked in three of the most significant hierarchies, visualized using multidimensional scaling.

## 5 CONCLUSION

We show here how dynamical equations alone are enough to create self-organizing systems capable of learning complex structures such as imbalanced and dynamical hierarchies. In fact, experiments have shown that these dynamical equations have two emerging properties that are typical of self-organization systems: (a) bottom-up organization and (b) presence of phase transition. Moreover, we propose Self-Organizing Dynamical Equations as a paradigm for machine learning together with an algorithm that implements it (Hierarchical TSFMap). Results show that, perhaps surprisingly, Hierarchical TSFMap is both more accurate and more adaptive than state-of-the-art algorithms in seven out of eight tasks.

This work also has implications in many areas such as cognitive science and neuroscience, shedding light on how self-organization circuits can be established as fundamental a mechanism in the brain. Results here suggest that the learning of chunking and hierarchical structures can be done by self-organizing circuits with Hebbian and anti-Hebbian plasticity. Thus, it reveals a relationship of Hebbian theory with brain self-organization and sets up the stage for novel cognitive theories to emerge, using self-organization as a principle rather than a byproduct.

## REPRODUCIBILITY STATEMENT

We have made the experiments easily reproducible by providing the following: (1) The parameters setting to reproduce results is available in Appendix. B. (2) The parameters setting and implementation details of our model are available in the main text and Appendix. A. (3) We have provided extensive details regarding the setup of the environment (generating input sequence) in Appendix. C. (4) Code to reproduce our environments from scratch, the proposed model, and the baselines will be submitted as supplementary material and made available through GitHub after acceptance. All the result for Imbalanced Hierarchical Structure and Dynamic Hierarchical Structure experiments was obtained from at least 30 independent experiments. Values in different experimental groups are expressed as the mean ± s.t.d. $p < 0.05$ was considered statistically significant. The p-values of each experiment are shown in Table 1.

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

## A HIERARCHICAL CHUNKING PHASE

The output reveals the information on how many levels of hierarchy can be distinctively identified and how variables of each level form chunks. We first perform linkage (Müllner, 2011) on input $w$ and return a distance matrix $Z$ (or sometimes referred as a linkage matrix). Using $Z$, we compute the distance between each formation of the non-singleton cluster as $\delta(Z)$. We index $\delta(Z)$ with ascending order, sort it according to branch distance with descending order $sort(\delta(Z)) = [d \in D | d_i > d_{i-1}]$ and return their corresponding index, where $sorted\_\delta_{id} = argsort(\delta(Z))$. In other words, index $sorted\_\delta_{id}$ represents the n-th number of the formation of the non-singleton cluster. We then remove the index in $sorted\_\delta_{id}$ that is larger than its previous index. The length of $sorted\_\delta_{id}$ defined as $H$ is considered as the total number of levels in the hierarchy. Lastly, We iterate over $sorted\_\delta_{id}$ and perform flat clustering from linkage matrix $Z$ based on the criteria of number of cluster $n\_cluster = n - sorted\_\delta_{id} - 1$ on the same flat level. Predicted chunks $y$ on each level are then combined to form the final output $Y$.

Doing so essentially prioritizes putting the chunks with the most distinctive distance feature together, and divisively decomposing them into smaller chunks while taking the overall distance feature of all chunks into account. This opposes merely performing hierarchical clustering as it does not infers the exact number of levels in the hierarchy and which levels are important enough to be highlighted (Bar-Joseph et al., 2001). Algorithm 1 displays the pseudo-code for this method. Though we feed Hierarchical TSFMap's weight as an input, other feature vectors are acceptable. Here we use linkage with single method (Gower & Ross, 1969), yet other methods such as complete, ward, or average can be used.

---

**Algorithm 1** Hierarchical Clustering Phase

---

$W \leftarrow$ *Retrieve Hierarchical TSFMap's weights*
$Z \leftarrow$ *Perform linkage on $W$ and return distance matrix*
$\delta(Z) \leftarrow$ *Return all the branch distance between each formation of non-singleton cluster*
$sorted\_\delta_{id} = argsort(\delta(Z))$ *Sort branch distance with descending order where $sort(\delta(Z)) = [d \in D | d_i > d_{i-1}]$ and return their corresponding index*
$maximum\_id \leftarrow sorted\_\delta_{id}[0]$
**for** $id \in sorted\_\delta_{id}$ **do**
    **if** $id > maximum\_id$ **then**
        $maximum\_id \leftarrow id$
    **else**
        *Remove $id$ from $sorted\_\delta_{id}$*
    **end if**
**end for**
*set $H$ with length of $sorted\_\delta_{id}$*
*Initialize predicted label $Y$*
**for** $h \in \{0...H\}$ **do**
    *Number of clusters $n_{cluster} = n - sorted\_\delta_{id}[h] - 1$*
    *Form clusters $y$ from linkage matrix $Z$ with the number of cluster $n\_cluster$ be the parameter*
    *Set $Y[h] = y$*
**end for**
*Return predicted label $Y$*

---

## B BASELINES

We compare Hierarchical TSFMap to Word2Vec Mikolov et al. (2013), Modularity Maximization Newman & Girvan (2004), SyncMap Vasconcellos Vargas & Asabuki (2021) and directly apply hierarchical chunking phase on a transition probability matrix of state on every experiment mentioned previously. NMI score, $NMI(\hat{Y}, Y) = 2\frac{I(\hat{Y};Y)}{H(\hat{Y})+H(Y)}$, is used as evaluation metric to compare predicted chunk $\hat{Y}$ with the true label $Y$ (provided by the environments) on each level of hierarchy, which produce the final score defined by $\frac{\sum_{i=0}^{\hat{L}} NMI(\hat{Y}_i, Y_i)}{\hat{L}}$. Where $I(\hat{Y}; Y)$ is the mutual information between $\hat{Y}$ and $Y$, $H(\cdot)$ is the entropy, $\hat{L}$ is the total number of hierarchy in the environments,

generated by the environments itself. Hierarchical TSFMap can produce more than $\hat{L}$ number of hierarchies $L$ in its matrix output $Y$. In this case, we only take the first $\hat{L}$ rows for evaluation as they represent the most distinctive hierarchies.

## B.1 SyncMap

SyncMap and Hierarchical TSFMap belong under the same learning paradigm - Self-Organizing Dynamical Equations. In summary, SyncMap learns by creating a dynamic map that performs chunking from sequence data. Here, we inherited the parameters setting from the previous work. Learning rate $\alpha$ is set to $1e - 3$. Input time delay $m$ is set to 10. We set the map dimension $k = 5$ to standardize with Hierarchical TSFMap's setting. Distance between weights is calculated using the Euclidean metric. Note that we replaced DBSCAN Schubert et al. (2017) with Hierarchical clustering in the clustering phase to remove the necessity to perform DBSCAN on different levels of hierarchies. SyncMap can identify well the global context of the variables. However, local context is usually difficult to extract due to the overlapping of local chunks. Moreover, little to no adaptation occurred in responding to the structural changes in the environment (Fig. 12).

## B.2 Word2vec

We adopted a Skip-gram Word2vec to compare with our model. The modified Word2vec used a dense deep neural network model that takes the shape of a Variational Autoencoder. The latent dimension is set to 3 and the output size is equal to the number of input sizes. The model is trained under 10 epochs with a batch size of 64, with a learning rate of $1e - 3$. A window of 100 steps was used to calculate the output probability of skip-gram. We then performed hierarchical chunking on the learned word embeddings to identify its hierarchical structure and chunks on each hierarchy. Since word embedding can encode items with identical features closer in a vector space, we assumed it might preserve the hierarchical relationships amongst variables to some extent. When inspecting Fig. 11, it is apparent that variables that share the same direct parent chunk are clustered together. Yet, the relationship of chunks beyond that is vague. This hints that Word2vec can learn the relationship of co-occurrence of local variables effectively, but can hardly preserve any global relationship amongst variables/chunks.

## B.3 Modularity Maximization

One of the community detection algorithms, Modularity Maximization is used here as a baseline. Using modularity as a measure, the modularity maximization approach is used to identify communities from a network. It begins with each node in its community and joins the pair of communities with maximum modularity until all nodes form into a single community. We can then select the maxima of modularity to decide on how to split the network into communities. With multiple local maxima, we constructed a hierarchy structure of communities Newman & Girvan (2004). Here, we used a modified Clauset-Newman-Moore Modularity Maximization algorithm to incorporate multiple local modularity maxima Clauset et al. (2004). Since modularity maximization can only operate under the premise of a graph, we transformed the sequence of inputs into an adjacency/transition probability matrix (Fig. 10), which can then turn into a weighted directed graph. Therefore, despite being a deterministic algorithm, an imprecise description of the adjacency matrix can induce uncertainty in the result. However, an accurate mean is achievable with a large sequence under the asymptotic central limit theorem. Although the input data, and therefore the problem, is different from the one seen in this work, sequence data and complex networks can interchangeably convert to one another (e.g., via an adjacency matrix from transition probabilities or a random-walk over a complex network). This reveals Hierarchical TSFMap's connection with complex networks. Having said that, the similarities stop here as both the objective and methodology differ.

## B.4 Hierarchical Chunking on Transition Probability Matrix

We recorded the occurrence of state to state from the input sequence and create the transition probability matrix of the current state to the next state (Fig. 10). We then applied Hierarchical Chunking Phase directly on the matrix as we considered it as a feature. The probability of the state transition does not necessarily reveal the hierarchical relationship among variables. Using the problems

in the imbalanced hierarchical structure environment, for example, variables within a bigger chunk have a smaller transition probability to other variables in the same chunk; Otherwise, if the chunk is smaller. This can induce an inaccurate cut-off on the dendrogram produced by hierarchical clustering and extract incorrect information about chunks. That being said, using merely the transition probability of state-to-state transition can not interpret the hierarchical relationship of variables. In the dynamic environment, TP matrix records all state transitions throughout the time step; it does not account for the changes occurring throughout the time step. Moreover, the accuracy of this method is also sensitive to the precision of the TP matrix itself.

## C  INPUT GENERATION

In our experiments, we consider the extraction of hierarchical structure and presume the input sequence comes in a discrete form. Overall, any events that can be serialized as a sequence can be processed by Hierarchical TSFMap. To reproduce the environments used in our experiment or to create a new environment, one can utilize a graph to generate sequence input. Such a graph $G = (V, E)$ resembles the distribution of variables, with $v \in V$ being a set of nodes and $e \in E$ being a set of edges. The graphs reveal the number of variables, the composition of chunks, and their hierarchical structure. A leaf node $v_l \in V_l$ signifies a variable; while a non-leaf node $v_c$ represents a chunk. A chunk is composed of sub-chunks or variables.

Based on $G$, we create an all-to-all connection weighted directed graph $T$ that describes the transition probability from variable to variable, composed of merely variables, with the weight of edges $\omega$ be the transition probability. The transition probability between variables $\omega$ can be defined as:

$$\omega = \frac{1}{[d(v_l, v_l')/2]^3} \tag{5}$$

with $d(v_l, v_l')$ be the path length between variables $v_l, v_l'$ in $G$. We then normalized all out-going edges from variables:

$$\hat{\omega}_{v_l, v_l'} = \frac{\omega_{v_l, v_l'}}{\sum^{|V_l|} \omega_{v_l, v_l'}} \tag{6}$$

that $\hat{\omega}_{v_l, v_l'} \in [0, 1]$. To generate a sequence of input, we apply a random-walker on graph $T$ to travel from variable to variable, based on the probabilities $\hat{\omega}_{v_l, v_l'}$ associated with the current variable. Variable is placed into the sequence input whenever random-walker travel to one.

In the imbalanced hierarchical structure and two real-world networks environments, the models receive 300,000 sequential input signals $S_t = \{S_1, S_2, \cdots, S_\tau\}$ where $\tau = 300000$. In the dynamic hierarchical structure environment, We doubled $\tau$ to 600,000 to accommodate the time step needed for adaptation. The number of variables, chunks, and hierarchies varies across different problems.

## D  EXPERIMENTAL SETUP FOR REAL-WORLD SCENARIOS

In this paper, we consider Zachary's karate club network and Lusseau's bottlenose dolphin social network as the modelings of real-world scenarios. Since the dataset we used here is graphs, we generate the input sequence using the same method from Appendix. C. For the experiments, ward linkage is used by all the baselines when performing the hierarchical chunking phase, except TP matrix, to increase the NMI score for all baselines.

The metric used to evaluate the performance of our model on this dataset can be difficult. Consider the case of the karate club network: despite the ground truth provided by the literature is usually two communities formed according to the factions; Members can form a smaller community within the faction. This indicated that such networks contain a hierarchical structure in them, which is useful as it provides more insight into the data but is often overlooked by literature. Hence, for evaluation: (1) we compare the predicted chunk from the most significant hierarchy with the ground truth provided using NMI score, and (2) visualize the representation formed in weight space/word embedding for the remaining hierarchies. Note that, we provided here the hierarchical structure extracted using

our model as a reference, rather than a ground truth, whereas the definition of it depends on various perspectives.

## D.1 ZACHARY'S KARATE CLUB NETWORK

The well-establish Zachary's karate club network data collected by Zachary (1977) represents the social interactions ties among the members of the club. Due to internal conflict, the club later split up into two factions that become the ground truth for the community detection/clustering of this dataset. This network contains 34 nodes and 78 edges, with nodes representing the members of the club and edges the presence of social interactions within or away from the karate club (Fig. 16).

## D.2 LUSSEAU'S BOTTLENOSE DOLPHIN SOCIAL NETWORK

Another famous network, Lusseau's bottlenose dolphin social network, is used as a benchmark to verify the performance of our model. The network contains 62 nodes that represent each individual bottlenose dolphin and 159 edges that represent the interaction between dolphin pairs observed to co-occur more often than expected. Furthermore, the ground truth of this network can be partitioned into two main groups (Lusseau et al., 2003); On the other hand, Cheng et al. (2014) considers the ground truth with four groups. After all, we compare our prediction to the former ground truth using NMI score while taking the latter as a reference when visualizing the representation formed in weight space (Fig. 17).

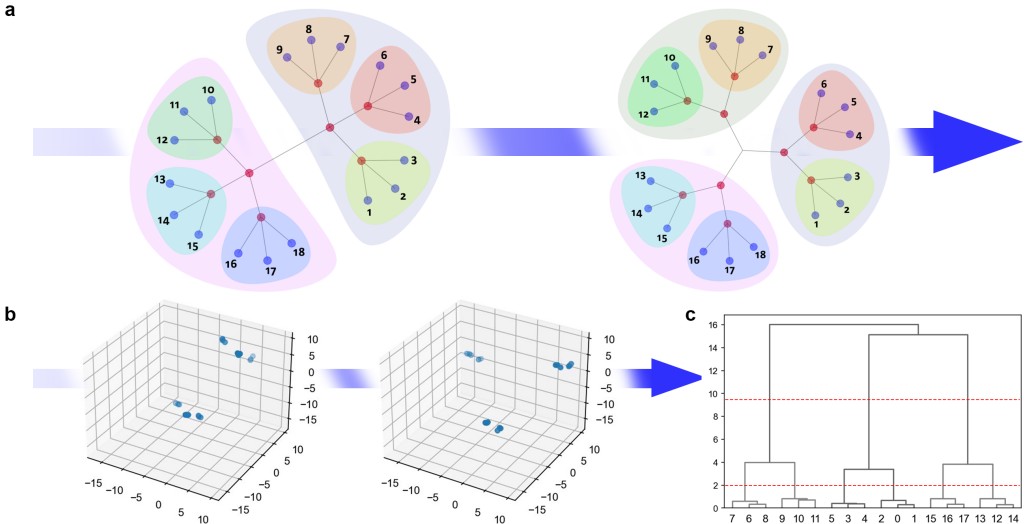

Figure 9: **Setting of dynamic hierarchical structure experiment.** (**a**) Leaf node indicates variables and node in red represents a chunk. Color coding is used to indicate chunks. Consequently, sibling nodes belong to the same chunk. A chunk can contain either child chunks or variables. The graph shows the distribution of how the sequence of variables is generated. Transition probability between variables increases as the number of edges connecting them decreases. Essentially, the transition probability is higher when variables belong to the same chunk. In a dynamical setting, the distribution of variables started with the graph on left, changes halfway through the time step to the graph on right. In this example, two big chunks are split into three big chunks. (**b**) The dynamic of weights in Hierarchical TSFMap reconfigure to adapt to the changes in the distribution of input sequence, in three-dimensional weight space. Observation shows that the weights first form into two big chunks and enter an equilibrium state. Once the distribution of the sequence is altered, the weights reconfigure their position to split into three big chunks. (**c**) The dendrogram produced during the hierarchical chunking phase, extracted from the weights in (**b**). The color coding reveals the hierarchical structure produced matches against the latest distribution of variables in (**a**).

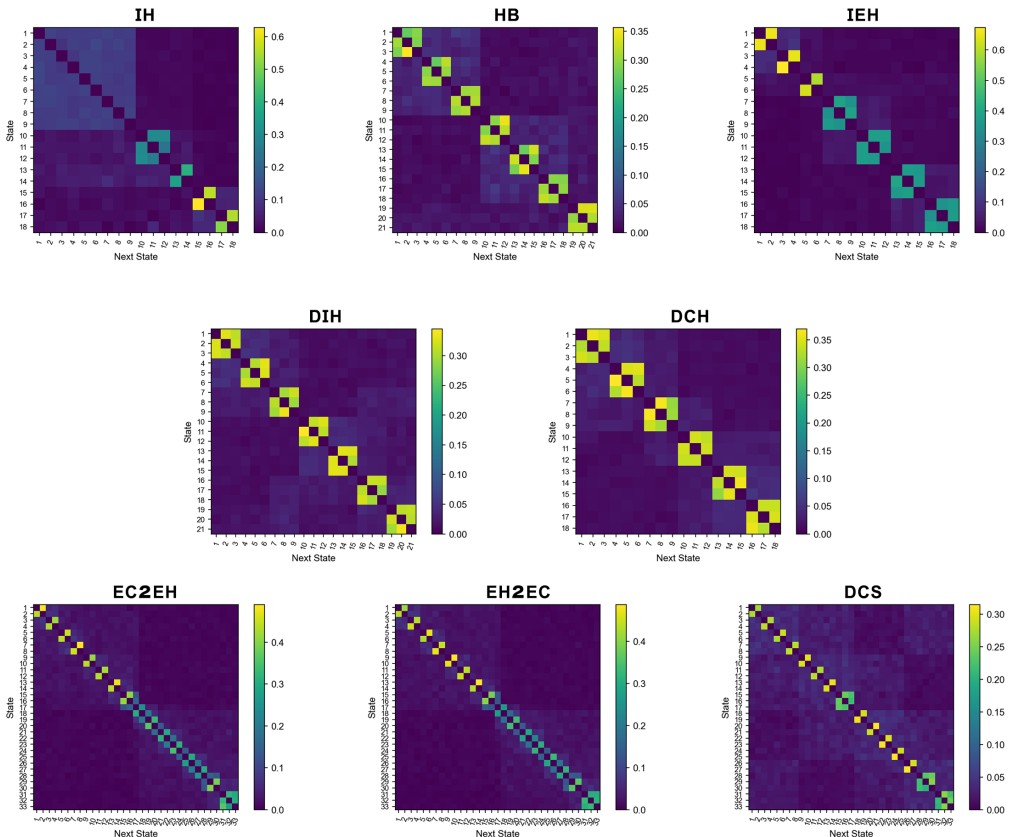

Figure 10: **Heat map of transition probability matrix of the current state (Y-axis) to the next state (X-axis).** (IH, HB, IEH) Transition probability matrix (TP matrix) from the imbalanced hierarchical structure environments. Alternatively, The recorded transition of state represents an adjacency matrix, convertible to an all-to-all connection weighted directed graph. The boundaries shown in the heat map usually hint at the presence of chunks. (DIH, DCH, EC2EH, EH2EC, DCS) TP matrix generated from dynamic hierarchical structure environments. It is important to note that the TP matrix has encoded the transition from two kinds of distribution. Therefore, the TP matrix failed to capture the dynamic of changes in environments. Consequently, baselines that utilized the TP matrix are only as accurate as the probability distribution is.

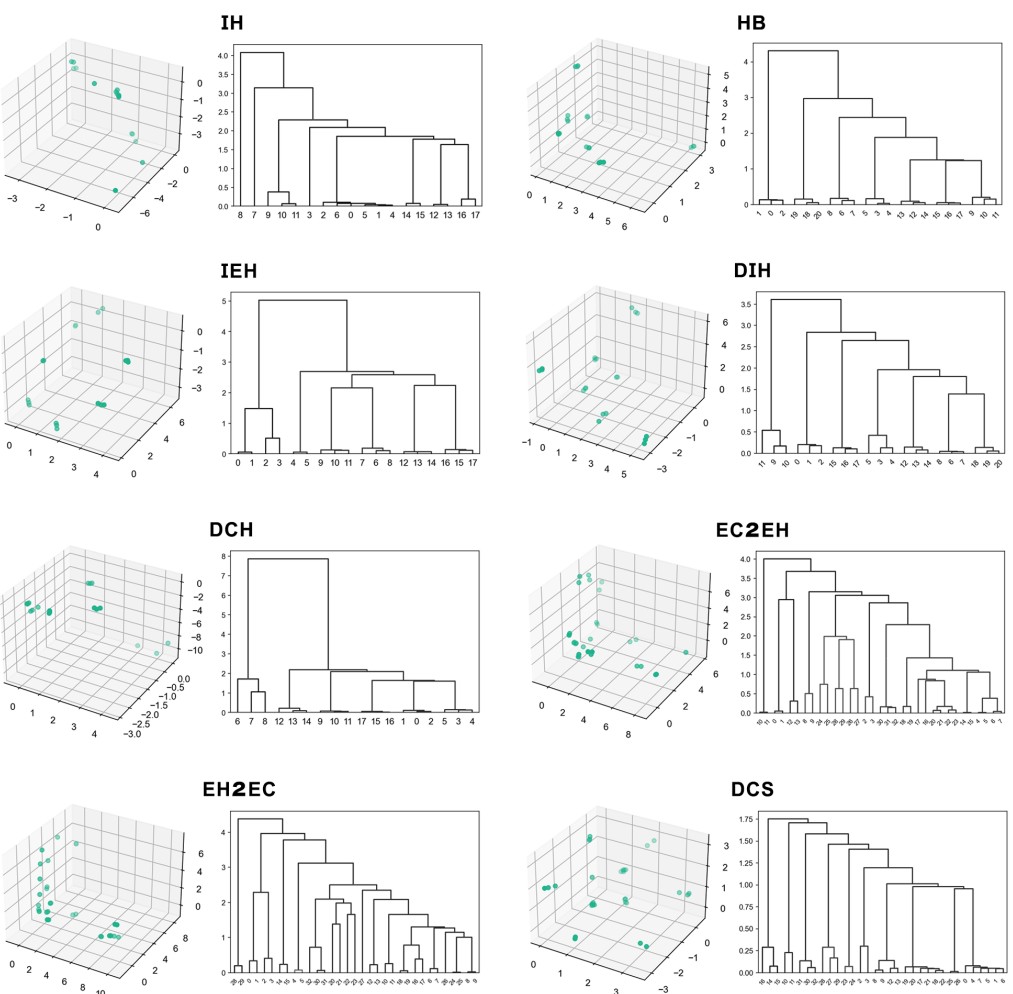

Figure 11: **Word embeddings of Word2vec.** The word embeddings of Word2vec are learned in a three-dimensional latent space and its hierarchical structure is produced by hierarchical clustering algorithm. This provides a clear indication that Word2vec can learn the relationship of co-occurrence of local variables effectively; however, the embeddings reveal that it hardly preserves the global relationship amongst variables.

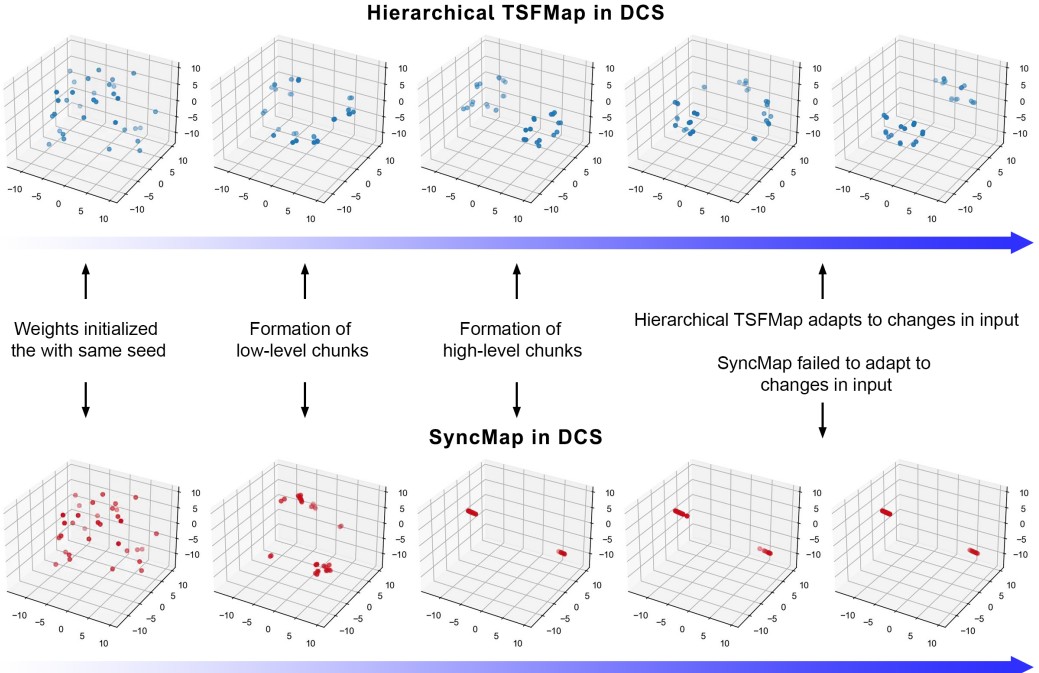

Figure 12: **Comparison of adaptive capability between Hierarchical TSFMap and SyncMap in DCS.** We observed that Hierarchical TSFMap is significantly better than SyncMap when adapting to the changes in the environment. To place both methods in a fair comparison, we initialized weights in a three-dimensional space with the same seed. Both methods managed to form hierarchies of chunks with respect to the input. Once the structural changes in the environment occurred, Hierarchical TSFMap adapt its weights and form a new pattern accordingly; in contrast, SyncMap failed to adapt and remained in the same configuration prior to the changes in the environment. Additionally, although initialized in a higher dimensional weight space (in this example, three-dimensional weight space), the rank of the weight matrix converged to 1 given enough time, $1 = \rho(w)$. This indicates that SyncMap's dynamic is restricted in one-dimensional space. The weight matrix of Hierarchical TSFMap however, can retain its high dimensionality, $1 \leq \rho(w) \leq k$.

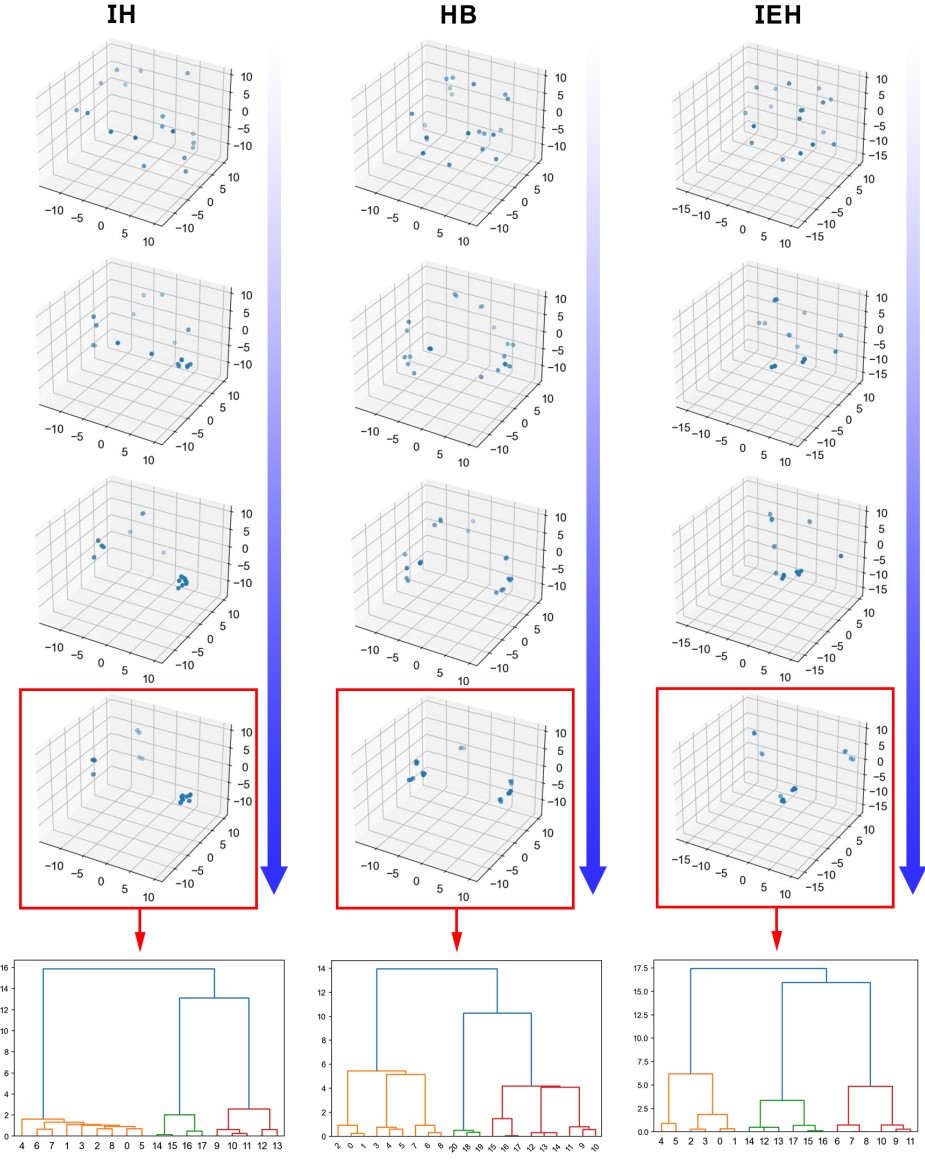

Figure 13: **The dynamics of Hierarchical TSFMap in Imbalanced Hierarchical Structure environments.** The dynamics start with a set of weights initialized in $\sigma$ space, and progressively form into pattern that corresponds to the input. The patterns are extracted to produce a dendrogram that describes the hierarchical structure of underlying data. Both the patterns and dendrogram portray somewhat distinctive hierarchical structures, even through manual inspection.

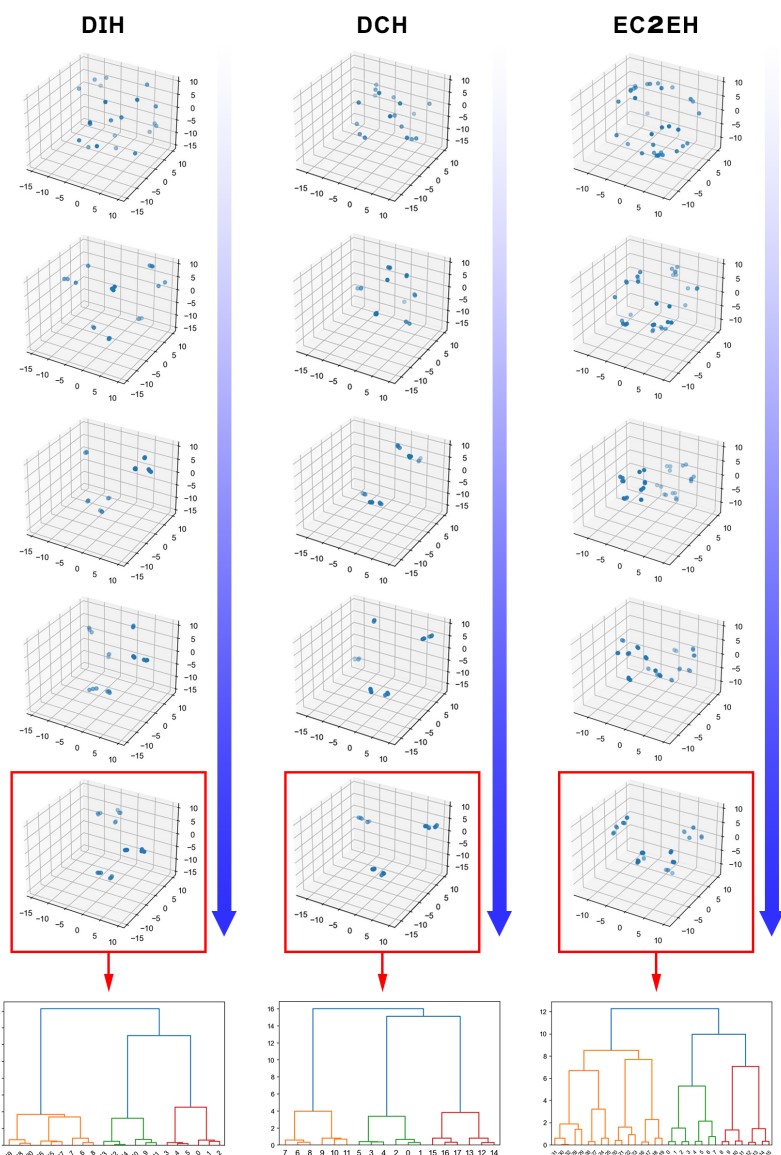

Figure 14: **The dynamics of Hierarchical TSFMap in Dynamic Hierarchical Structure environments.** Notice that the $\sigma$ space shown in the third and fifth rows are the patterns corresponding to the first and second data distribution respectively. The latest pattern formed is used for identifying the hierarchical structure of the latest changes in data distribution. This proves that, given enough time, Hierarchical TSFMap can adapt to any distribution of data, regardless of the current state; for example, from either random initialization or pattern that are already established.

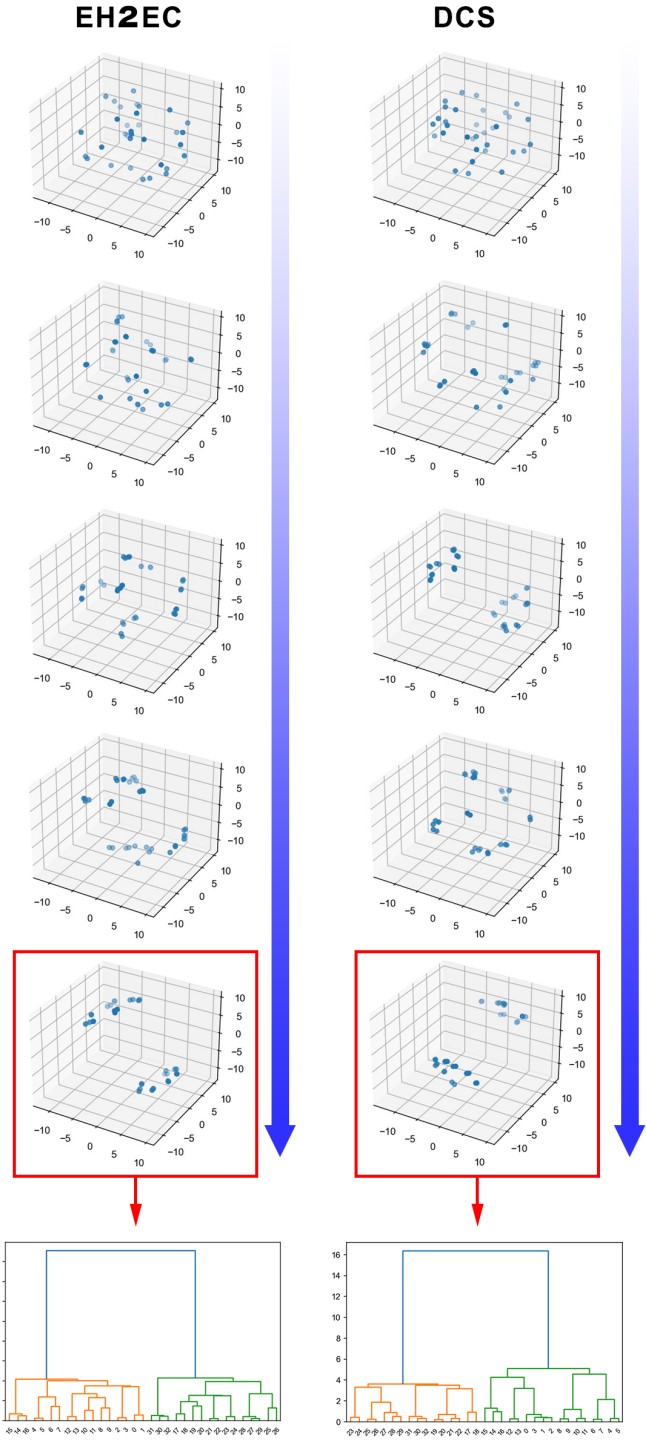

Figure 15: **Refers to Fig. 14.**

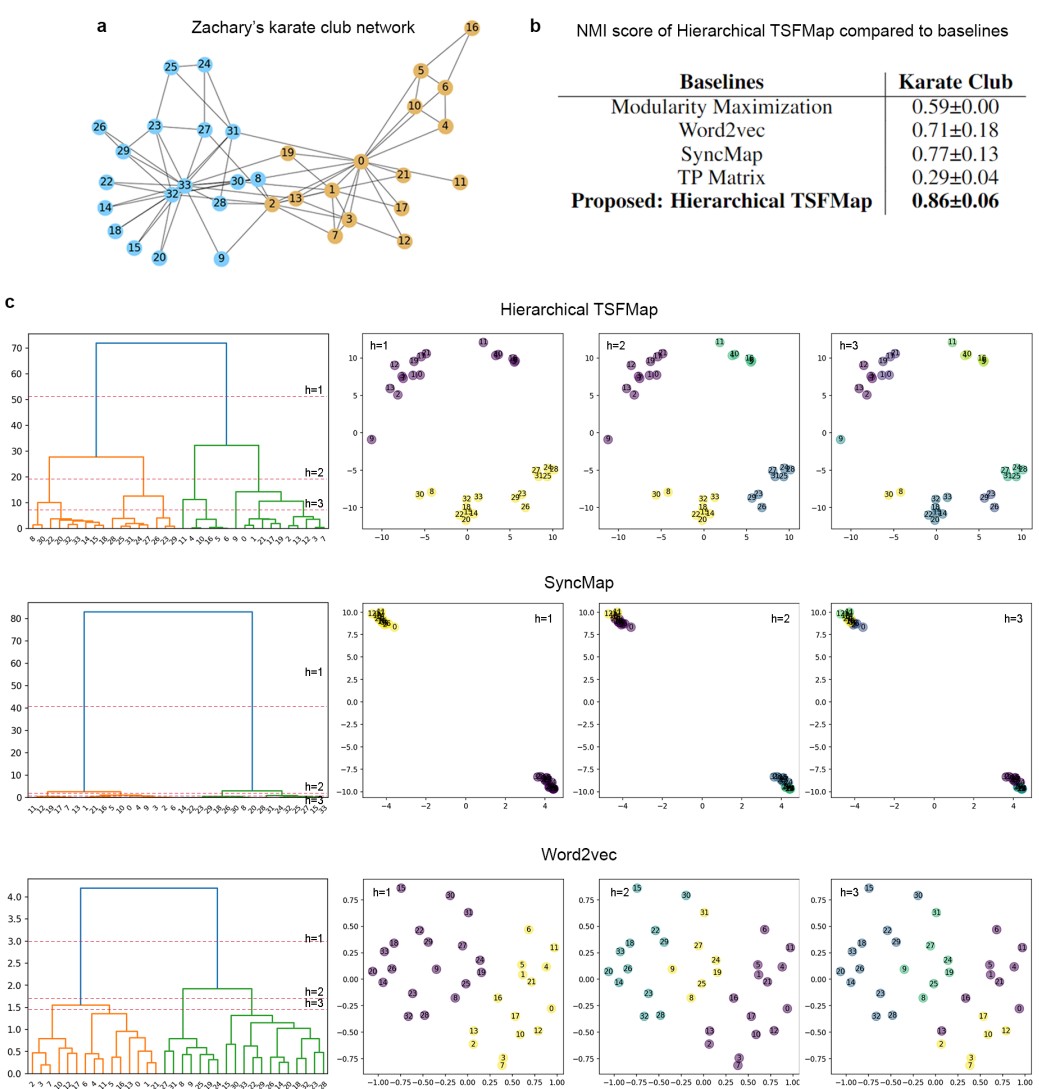

Figure 16: Extracting the hierarchical structure of Zachary's karate club network. (a) The ground truth of the Karate club network provided by Zachary (1977). (b) The NMI score of Hierarchical TSFMap compared to its baselines, comparing only predicted chunks from the most significant level of hierarchy to the ground truth. (c) The weight space/word embedding of Hierarchical TSFMap, SyncMap, and Word2vec. The weight space ($k = 5$) describes how weights are chunked in three of the most significant hierarchies, visualized using multidimensional scaling. The predicted chunks are labeled by color. The dendrogram shows the hierarchical clustering of weights, where the red dashed lines are the cut-off. It appears that the models shown here can configure their weights to match the ground truth in most instances, with Hierarchical TSFMap being the most accurate one reflected by the relatively high NMI score in (b). Interestingly, members can form a smaller community within the faction. Removal of hubs can further break down a faction into a smaller community and exhibit the property of hierarchy. It is also demonstrated in the weight space of Hierarchical TSFMap, when looking deeper into the hierarchy, smaller groups of members and less social individuals can be formed into their own chunks. Yet, a similar occurrence is not displayed in the weight space/word embedding of SyncMap/Word2vec. Note that, individuals (i.e., node 9) that are weakly associated with both factions cannot be mapped in the middle of the two factions. This is due to the fact that edges are not weighted and hubs with smaller connectivity have a larger probability to attract them.

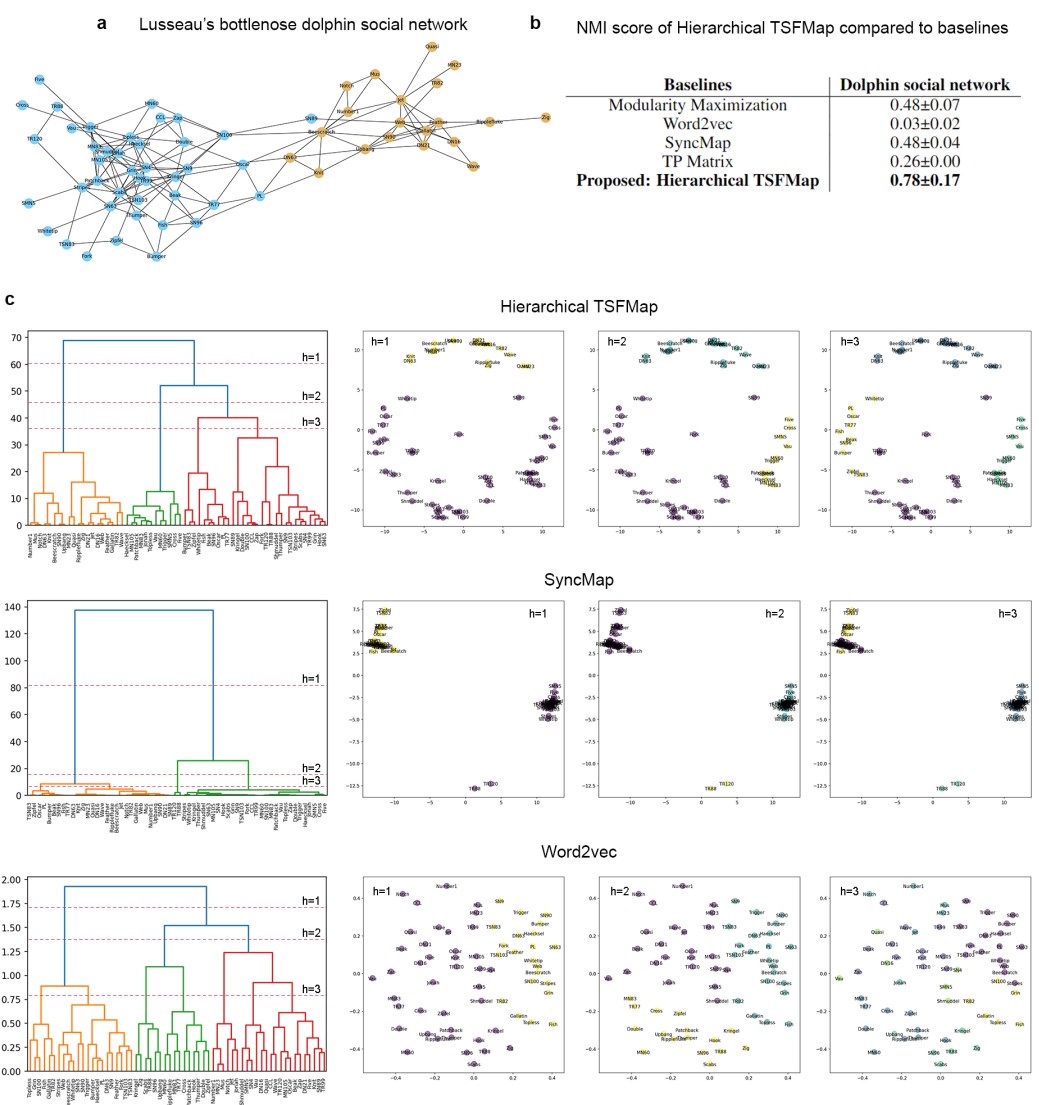

Figure 17: Extracting the hierarchical structure of Lusseau's bottlenose dolphin social network. (a) The ground truth of the dolphin social network provided by Lusseau et al. (2003). (b) The NMI score of Hierarchical TSFMap compared to its baselines, comparing only predicted chunks from the most significant level of hierarchy to the ground truth. (c) The weight space/word embedding of Hierarchical TSFMap, SyncMap, and Word2vec. The weight space ($k = 5$) describes how weights are chunked in three of the most significant hierarchies, visualized using multidimensional scaling. The predicted chunks are labeled by color. The dendrogram shows the hierarchical clustering of weights, where the red dashed lines are the cut-off. We observed that Hierarchical TSFMap can form chunks on the most significant level of the hierarchy accurately when compare to the ground truth. More surprisingly, going deeper into the hierarchy, the weights can form into four groups, similar to the ground truth provided by Cheng et al. (2014) with slight differences. On the other hand, SyncMap can form two chunks with lower accuracy, whereas some nodes are incorrectly placed. Lastly, Word2vec failed to learn meaningful chunks, hence the lower NMI score.

Table 1: Statistical Results. We used a two-sample t-test with a p-value larger than 0.05 to accept the null hypothesis that the means are equal. We calculate the two-tailed $p$ value. The results were obtained from at least 30 independent experiments.

| Environments | SyncMap | Hierarchical TSFMap | Word2vec | Modularity Maximization |
|---|---|---|---|---|
| IH | 0.424 | 0.518 | 0.854 | 1.000 |
| HB | 0.618 | 0.766 | 0.496 | 1.000 |
| IEH | 0.846 | 0.153 | 0.734 | 1.000 |
| DIH | 0.535 | 0.673 | 0.635 | 1.000 |
| DCH | 0.578 | 0.556 | 0.932 | 1.000 |
| EC2EH | 0.897 | 0.173 | 0.649 | 1.000 |
| EH2EC | 0.554 | 0.368 | 0.395 | 1.000 |
| DCS | 0.880 | 0.759 | 0.639 | 1.000 |
| Karate club network | 0.882 | 1.000 | 0.553 | 0.440 |
| Dolphin social network | 0.629 | 0.702 | 0.541 | 0.362 |

