# OpenReview forum: "Dynamical Equations With Bottom-up Self-Organizing Properties Learn Accurate Dynamical Hierarchies Without Any Loss Function"
_ICLR.cc/2023/Conference — Submitted to ICLR 2023_

### Official Review · Reviewer_iaFP · 2022-10-24

**Confidence:** 3
**Correctness:** 3
**Technical Novelty And Significance:** 3
**Empirical Novelty And Significance:** 2
**Recommendation:** 5

**Clarity, Quality, Novelty And Reproducibility:**

To the reviewer's knowledge, the present work represents a novel approach to ascertaining hierarchical structure in data, one that explicitly allows for an online inference of dynamically changing inputs.

Unfortunately, certain aspects are left unclear and hinder reproducibility:

1. I found the input encoding difficult to parse.  This could be improved with a clearer explanation and by using Figure 2a to clarify the relevant variables ($x_i, S_t, s_{i,t}, n, k ,\tau, ...$)

2.  The authors refer to the $\sigma$ space of their method (e.g., Fig 2c), but I was unable to find this defined anywhere in the manuscript. What is it, and what determines its dimensionality?

3. $k$ is given as a hyperparameter and set to 5 for most of the analyses, but it is unclear why this value is used.  What happens if different values of $k$ are used? Is there a way to infer what the value of $k$ should be?

4. The 2nd term of the inactive portion of Eq. 3 has a divisor of $d^2$ rather than $d$ like the other terms have.  Is there a reason for this asymmetry to be introduced?  What happens if $|d|$ is used instead, e.g.?

5. What are the input sequences used for the results shown in Figs 4-7?  While having this provided in code will be useful, the manuscript's readability would be enhanced by including this information.

6. While p < 0.05 is considered signficant, no p-values are provided, hindering the interpretation of the results

**Strength And Weaknesses:**

Strengths:

The authors provide a novel, yet simple dynamical method that infers the hierarchical structure of dynamical inputs well. The results appear to improve upon current methods and may provide insight into how biological neural networks perform similar analyses on incoming data streams.

Weaknesses:

The manuscript would benefit from a clearer explanation of the method and from the example inputs that were used being made explicit (please see below for more details).

**Summary Of The Paper:**

The authors have introduced a dynamical method to infer the hierarchical structure from temporal sequences. Inputs are encoded into temporal sequences of exponentially-decaying spikes that then determine the temporal evolution of weight variables associated with the inputs. Clustering is then performed to determine the hierarchical structure encoded in the weights.  The authors show that their method outperforms other hierarchical methods when used on imbalanced or dynamically changing hierarchies.  To the reviewer's knowledge, this work represents a novel method of dynamically learning and inferring hierarchical structure in a bottom-up manner from sequential data.

**Summary Of The Review:**

The present work provides a novel method to dynamically learn and infer hierarchical structures from sequential data that appears to perform favorably compared to other methods.  However, the manuscript is hampered by a lack of clarity in several aspects.

---

> ### Author Response · Authors · 2022-11-16
> **Rebuttal**
>
> We thank the reviewer for their helpful feedback and a comprehensive summarization of our paper. The reviewer’s description of our paper clearly captures the essence of our idea. Below, we have responded to the feedback from the review.
>
> > I found the input encoding difficult to parse. This could be improved with a clearer explanation and by using Figure 2a to clarify the relevant variables $(x_i,S_t,s_i,t,n,k,τ,...)$
> >
>
> This is a good suggestion! We added the notations to Figure 2 to better complement the readability of our paper.
>
> > The authors refer to the $\sigma$ space of their method (e.g., Fig 2c), but I was unable to find this defined anywhere in the manuscript. What is it, and what determines its dimensionality?
> >
>
> Thank you for pointing that out. We added a definition of the $\sigma$ space in section 3. It is the space that holds weights, allowing weights to maneuver and form patterns. Its dimensionality is determined by the hyperparameter $k$.
>
> > $k$ is given as a hyperparameter and set to 5 for most of the analyses, but it is unclear why this value is used. What happens if different values of $k$ are used? Is there a way to infer what the value of $k$ should be?
> >
>
> Many values were tried when experimenting with how the dimension of weight space $k$ can affect the performance of our model. We observed that $k > 5$ can increase the learning efficiency but not by a large margin, hence 5 is sufficient for the setting of our problems. Generally speaking, the higher the dimension, the higher the degree of freedom for weights to “maneuver” in the weight space; Therefore, the faster the weights can enter an equilibrium state. Also, if the dimension of the weight space is low, the information on the correlation between chunks can be lost. For example: if $k = 1$, the distance can only account for the accurate correlation between two chunks. The inference of a preferred value of $k$ is simply akin to the hyperparameters setting for the width and height of a neural network, which depends on the trade-off between efficiency and computational cost, as long as the problem is tractable.
>
> > The 2nd term of the inactive portion of Eq. 3 has a divisor of d2 rather than d like the other terms have. Is there a reason for this asymmetry to be introduced? What happens if $|d|$ is used instead, e.g.?
> >
>
> The term $F3$ acts as a repellent force that repels inactive states away from $cp$. That being said, a large $F3$ can break the formation of chunks that are already formed. Therefore, $d^2$ act as a regularization term that exponentially reduces the force as the inactive states get further away. This can actively help uncorrelated states to maneuver around other irrelevant chunks. at the same time, keeping the formation of existing chunks. Since $d$ (the distance of weights away from $cp$) will always be positive, $|d|$ is equivalent to $d$.
>
> > What are the input sequences used for the results shown in Figs 4-7? While having this provided in code will be useful, the manuscript's readability would be enhanced by including this information.
> >
>
> The details of generating the input sequences used in the mentioned experiments are mentioned in the appendix section C. In short, the sequences are generated by applying a random-walker that travels on the graphs in Fig. 4, 6, and 7. Nodes within the same chunk have a higher probability of transitioning to each other. With that, the input sequences also contain a hierarchical structure in it. Nonetheless, we follow the reviewer’s suggestion and added a shorter description of the information from section C to make the manuscript more readable, highlighted in section 4. Additionally, we created a short movie as supplementary material to illustrate the input sequence generation process.
>
> > While p < 0.05 is considered signficant, no p-values are provided, hindering the interpretation of the results
> >
>
> We agree with this statement. Therefore, we added a new section/table in the appendix to provide the p-values for all the experiments. In fact, the code for the statistical test (including the one that outputs the p-values after running the experiments) is provided.

---

> > ### Comment · Reviewer_iaFP · 2022-12-06
> > **The authors have addressed some of my concerns, but others remain**
> >
> > I thank the authors for their responses.  Overall, the responses and modifications have addressed aspects of the clarity I had questions about.  However, I don't believe the authors have fully addressed other concerns that have been brought up.  In particular, as reviewer fCck pointed out, the present work (1) does not clearly compare against state-of-the-art methods, and (2) appears to represent a somewhat minor conceptual advance over and above the SyncMap model from the cited 2021 AAAI work.  Moreover, the reported $p$-values in Table 1 also present some cause for concern.
> >
> > _**Comparisons**_
> >
> > In particular, for (1), while the authors offer the successful Word2Vec embedding scheme as an unsupervised clustering analog, BERT--referenced by reviewer fCck--represents a more advanced and state-of-the-art form of such an embedding model.  It is unclear how well BERT or other, more up-to-date, self-supervised embeddings might perform in the tasks outlined by the authors.
> >
> > _**Contribution**_
> >
> > Regarding (2), it appears that the authors' present work mostly involves using the framework outlined in the 2021 AAAI paper and adding a hierarchical phase at the end, the "Hierarchical Chunking Phase" described in Appendix A.  If this is indeed, roughly, the case, then it seems as though this should be made more clear by the authors.  That is, the authors could better describe their contribution as helping to solve hierarchical, dynamic clustering in a manner that might be more biologically plausible via a novel combination of methods involving a recently-introduced discrete dynamical system that more plausibly models neuronal responses to inputs with other (known in part at least), unsupervised hierarchy-discovery algorithms.
> >
> > _**Hierarchical Chunking Phase**_
> >
> > Such a description would avoid some of the connections to neuroscience and psychology that the authors have advanced, such as linking the present work to theories of top-down learning, and that may be somewhat beyond the scope of the dynamical hierarchical results that are provided. In that case, it appears as though the Hierarchical Chunking Phase (HCP) should not be left to an appendix.  Furthermore, I unfortunately am a bit uncertain as to the workings of the HCP algorithm as outlined. I overall find the description and pseudocode difficult to parse (e.g., it seems to delete all sorted indices that are not maximal, leaving one element, yet then make a loop based on the number of elements of the sorted indices).
> >
> > **_p-values_**
> >
> > Finally, related to the HCP algorithm, I have concerns about how the $p$-values that the authors have added in Table 1 are used. If I understand the table correctly, the authors do not have any statistically significant results.  I'm additionally concerned about the manner in which they use statistical significance. The authors have statements such as "The weight space ($k = 5$) describes how weights
> > are chunked in three of the most significant hierarchies" in Fig 17 ("most significant" is a phrase that appears 9 times).  However, it appears that the "most significant" $p$-values, based on Table 1, are still far from being statistically significant (since the Table does not break the results into layers, I assume the reported $p$-values are the ones that are closest to 0).
> >
> > **_TL;DR_**
> >
> > In the end, it appears that the core novelty is the combination of known dynamical model + HCP algorithm (aspects of which might be novel).  Yet, I'm left unclear about exactly how the HCP algorithm works and is reliant on the $p$-value (or its ranking).  However, it appears as though a non-traditional interpretation of $p$-values might, worryingly, strongly influence the reported results.
> >
> > In light of these difficulties, I will leave my score at 5.

---

### Official Review · Reviewer_S42c · 2022-10-25

**Confidence:** 2
**Clarity, Quality, Novelty And Reproducibility:** The paper could definitely be clearer.
**Correctness:** 4
**Technical Novelty And Significance:** 2
**Empirical Novelty And Significance:** 3
**Recommendation:** 3

**Strength And Weaknesses:**

Strengths:
The paper proposes a method for exploring sequence data and extracts a graphical representation of possible interaction among features/variables, as the

Weaknesses:
The use of ‘dynamical equations’ in the title could create a wrong impression. It seems to be about $w_i$ update that about dynamics in the sequence data.
The setting of the dynamic data generation is confusing to me: which part of it is real data-based and what is synthetic?
It is not clear what one would achieve with the final network/networks outputted by the algorithm.


Minor issues:
The explanation of the term associated with ‘non-recently activated input set’ in equation (3), in text in the paragraph after equation (4), has some extra minus signs.



**Summary Of The Paper:**

This paper is a method for finding embedding for components of encoded dynamical data that captures “co-occurance” of state variables and then perform hierarchical clustering on those embeddings. This method is compared to other representations.

**Summary Of The Review:**

Even if there are some interesting ideas in this paper, it is hard to get very excited about another clever exploration of complex network data.

---

> ### Author Response · Authors · 2022-11-16
> **Rebuttal**
>
> Thank you for reviewing the paper. We have responded below to the individual feedback from your review.
>
> > Strengths: The paper proposes a method for exploring sequence data and extracts a graphical representation of possible interaction among features/variables, as the
> >
>
> Thank you for mentioning the strength of our paper. However, there is an incorrect description of our method. Our model extracts hierarchical information represented by the distance between weights, rather than a graphical representation. Also, some feedback seems to be missing as the sentence is incomplete.
>
> > The use of ‘dynamical equations’ in the title could create a wrong impression. It seems to be about $w_i$ update that about dynamics in the sequence data.
> >
>
> We believe this is another misinterpretation of our method. The weights $w_i$ do not update the dynamics or the sequence data; Instead, the dynamical equations update the $w_i$ to represent the temporal correlation of the input variables in sequence data.
>
> > The setting of the dynamic data generation is confusing to me: which part of it is real data-based and what is synthetic?
> >
>
> The sequence data used in the experiments for sections 4.1 and 4.2 are synthetic. Besides, the sequence data used in section 4.3 came from the real-world dataset. The setting of data generation is included in appendix section C. We apologize for the lack of description in the main text. To improve this, we have added a shorter description of data generation in section 4.
>
> > It is not clear what one would achieve with the final network/networks outputted by the algorithm.
> >
>
> The final output of our algorithm is not a network. Hierarchical TSFMap outputs the predicted chunk label for each level of the hierarchy. In detail, it is an $L × N$ matrix,  with $L$ being the total levels of hierarchy and $N$ being the number of input variables. The final output is useful for showing the correlation of the input variables on different levels of the hierarchy.
>
> > The explanation of the term associated with ‘non-recently activated input set’ in equation (3), in text in the paragraph after equation (4), has some extra minus signs.
> >
>
> Thank you for pointing it out. We have fixed the typo in the revised paper.
>
> > Even if there are some interesting ideas in this paper, it is hard to get very excited about another clever exploration of complex network data.
> >
>
> The exploration of complex network data is one of the demonstrations of real-world applications for Hierarchical TSFMap. The reason complex network data is used is that the hierarchical structure of a complex network is more apparent. In the end, any dataset that can be serialized as sequential data can be extracted with our model if there exist different chunks and hierarchical structures in the dataset.

---

### Official Review · Reviewer_fCck · 2022-10-25

**Confidence:** 4
**Correctness:** 3
**Technical Novelty And Significance:** 2
**Empirical Novelty And Significance:** 3
**Recommendation:** 5

**Clarity, Quality, Novelty And Reproducibility:**

- Clarity is fair
- Novelty is not major; no theoretical study or measurement on the mutual information bound.

**Strength And Weaknesses:**

Pros

- The authors extend the dynamic equation in the complex network community for a community detection problem.

Cons

- Overall, the current application is still limited to a narrow problem.

There are also some neuro-circuit and gene-circuit are much often serving as public benchmarks.

- The framework is very incremental from the previous study [AAAI 2021].

- the baselines are weak and out of date. Some graph embedding and more recent word embedding methods are missing.

**Summary Of The Paper:**

- Overview

This work studies how to introduce dynamic equations for the complex networks under self-organization extended from previous work on [AAAI 2021].

The authors introduce a method called Hierarchical Temporal Spatial Feature Map (TSFMap) that uses a weighted coefficient learning manner to encode input sequences.

The workflow further combines with the chunking phase to cluster data. A related dynamical equation has also been studied in the design of gene circuits [Ref 1].

- Baseline

The authors only use word2vec (2013) to serve as one representation of learning, when several methods, such as GloVe, FastText, BERT, are actually much more widely adapted baselines in the representation learning community.

The authors studied simulated data, karate club network, and bottlenose dolphin social network data in the real-world and evaluated its mutual information scores.

**References**

1. Controllability, multiplexing, and transfer learning in networks using evolutionary learning, 2018
2. Glove: Global vectors for word representation, EMNLP 2014
3. Enriching word vectors with subword information, TACL 2017
4. FastText.zip: Compressing text classification models, 2016


**Summary Of The Review:**

Overall, this work tackles a problem with the limited interest in the representation learning community with weak baselines.

Furthermore, that could be a plus if the authors conduct some experiments on the information flow in some common biological, but it is unfortunately missing.


### Post-rebuttal

Thanks author for the efforts on some baselines. However, the nlp baselines are still very weak. (e.g., Glove was proposed in 2014)

With both experience in the complex system and language modeling, I did not see a major benefit from the proposed method.

I consider this paper with limited interests in a very narrow problem. This is a 4 score paper to me.

---

> ### Author Response · Authors · 2022-11-16
> **Rebuttal**
>
> Thank you for reviewing the paper and suggestions toward the baseline. We have responded below to the individual feedback from your review.
>
> > The authors extend the dynamic equation in the complex network community for a community detection problem.
> >
>
> Despite using community detection problems as a demonstration of real-world application, it is important to know that our method is mainly used to handle sequential input/time series data. any serializable data, such as sequences of objects/events can be adopted as input.
>
> > The workflow further combines with the chunking phase to cluster data. A related dynamical equation has also been studied in the design of gene circuits [Ref 1].
> >
>
> Thank you for recommending an interesting reference that utilized dynamic equations to achieve learning, at the same time also showing the ability to be adaptive (transfer learning), which is one of the aims of our model. The difference lies in that the set of dynamic equations controls the dynamics of node/network structure, while the learning can be achieved partially by other learning algorithms compared in the paper. Our model, however, achieved learning entirely via the dynamical equations. Taking the reviewer’s recommendation, we have included this work as part of the related works in the revised paper.
>
> > The authors only use word2vec (2013) to serve as one representation of learning, when several methods, such as GloVe, FastText, BERT, are actually much more widely adapted baselines in the representation learning community.
> >
>
> > the baselines are weak and out of date. Some graph embedding and more recent word embedding methods are missing.
> >
>
> We agree with the reviewer that our baseline for word embedding methods is outdated. Therefore, we performed an additional comparison with one of the methods (GloVe) recommended by the reviewer and have will include the result on top of the original result in the revised paper. Please refer to the rebuttal summary for details.
>
> > Novelty is not major; no theoretical study or measurement on the mutual information bound.
> >
>
> The novelty of our paper lies in what we have described in the rebuttal summary. Regarding the mutual information bound, we derived the NMI lower bound by:
>
> $NMI(\hat{Y}; Y) \ge HCP \left (log \frac{t D(x_i)}{a} \right )$
>
> Where $HCP$ is denoted as the Hierarchical Chunking Phase function and $D$ as the dynamical equations Eq. 3 and 4. The lower bound is proportional to the t given for adaptation and the learning rate. In essence, when the learning rate is large, the time needed for adaptation becomes lesser, but the lower bound decreased as the stability of the system increased. We are consulting with other experts and will include this analysis in the revised paper once it is refined. Nevertheless, reviewer 7qEB raised the same concern with regard to theoretical study, where our response to the reviewer can also be applied here.
>
> > Furthermore, that could be a plus if the authors conduct some experiments on the information flow in some common biological, but it is unfortunately missing.
> >
>
> This will be an interesting application of the model! Unfortunately, the experiment is beyond our field without collaboration with experts from the field. Experimenting on complex network data, although simple, is easier to visualize and interpret when introducing our method.

---

### Official Review · Reviewer_7qEB · 2022-10-27

**Confidence:** 2
**Correctness:** 3
**Technical Novelty And Significance:** 3
**Empirical Novelty And Significance:** 3
**Recommendation:** 8

**Clarity, Quality, Novelty And Reproducibility:**

Clarity:
* the paper is written clearly.


Quality:
I am not an expert on pattern forming, so I can't evaluate if the proposed method is of high quality or novel.

Reproducibility:
the code is provided, so the main claims seem reproducible.

**Strength And Weaknesses:**

Strength:
* seems a novel unsupervised learning method

Weakness:
* Can the dynamics be written in the form of a differential equation? That might help with a more rigorous theoretical analysis of the dynamics of the TSFMap. Given that the link to pattern formation is stressed so much in the manuscript, an analytical analysis with the rich tool set of the field of Pattern Formation and Dynamics in Nonequilibrium Systems ( https://doi.org/10.1017/CBO9780511627200 ) would strengthen the submission.
* Both abstract, introduction and conclusion should be more specific about the innovation here which (as far as I understand) is to cluster data with hierarchical structure. Some statement in the paper about the novel hierarchical clustering method seem overly generic and slightly more pompous than the actual result (e.g. "This work also has implications in many areas such as cognitive science and neuroscience, shedding light on how self-organization circuits can be established as fundamental a mechanism in the brain.", "In this paper, we show how the learning of patterns can be achieved by Hebbian and anti-Hebbian learning dynamics, linking between Hebbian learning and top-down theories of intelligence", "Our proposed system is arguably more biologically plausible, and it is also shown to be more
accurate and adaptive than state-of-the-art unsupervised algorithms. In fact, it sets up a foundation for a new paradigm in machine learning solely based on self-organization from dynamical equations, namely Self-Organizing Dynamical Equations, which are inherently accurate and adaptive."). Are more clear-cut and less pretentious presentation might help readers to appreciate the actual contribution of the paper.

**Summary Of The Paper:**

The submission proposes a novel unsupervised learning method for dynamically identifying hierarchical structures called "Hierarchical TSFMap". The method is based on three parts, encoding, dynamics and a 'hierarchical chunking phase'. During the encoding, sequential data is encoded via temporally exponentially decaying kernels. In the dynamics, pattern forming occurs via the TSFMap.

**Summary Of The Review:**

The submission suggests a novel method for unsupervised identification of hierarchical structures in sequential data. While there was extensive numerical evaluation, a theoretical analysis of TSFMap was missing and should be added. E.g. the map could be written as a PDE and a link to the extensive literature on pattern formation in PDEs could be made, existence and uniqueness of solutions should be discussed. Note that I am not an expert on unsupervised learning/pattern formation.

---

> ### Author Response · Authors · 2022-11-16
> **Rebuttal**
>
> Thank you for the thorough review and the kind words regarding clarity and reproducibility. We have responded below to the individual feedback from your review.
>
> > Can the dynamics be written in the form of a differential equation? That might help with a more rigorous theoretical analysis of the dynamics of the TSFMap.
> >
>
> > While there was extensive numerical evaluation, a theoretical analysis of TSFMap was missing and should be added. For E.g. the map could be written as a PDE and a link to the extensive literature on pattern formation in PDEs could be made, existence and uniqueness of solutions should be discussed.
> >
>
> We agree that the theoretical analysis can help understand the dynamics of TSFMap. However, we wonder if it can be converted into PDE form by the complexity of some of the functions (e.g., indicator functions, group functions) in the dynamical equations, as it might not be analytically tractable. Previously, we expect the theoretical analysis to be beyond the scope of this paper. Based on the feedback,  intend to further expand on the analysis as discussed in the revised paper, with time permitting.
>
> > Reproducibility: the code is provided, so the main claims seem reproducible (However, I wasn't able to reproduce the results myself because one of the required libraries was missing on my end).
> >
>
> In order to make sure of its reproducibility, we have uploaded the Conda environment file to ease the process. Also, please tell us which required libraries are missing on your end.

---

> > ### Comment · Reviewer_7qEB · 2022-11-24
> > **acknowledgements of updates and code**
> >
> > We thank the authors for the revisions and update to the code. While I am not an expert on unsupervised learning/pattern formation, it seems that major objections of other reviewers were addressed. Therefore, I would give the submission the benefit of the doubt, thus I have slightly improved my recommendation.

---

### Author Response · Authors · 2022-11-16
**Rebuttal Summary**

We thank all the reviewers for their constructive feedback and suggestions. Based on the feedback, we have added several improvements and corrections to our paper. The additional improvements are summarized below:

- We have uploaded a movie as supplementary material to support our paper. This movie showed (1) the dynamic of TSFMap, (2) the data generation process, (3) the properties of TSFMap, (4) the workflow of TSFMap, and (5) a comparison of the learned map with Word2vec and SyncMap.
- The Conda environment file (env.yml) is uploaded as supplementary material to better ensure the reproducibility of our work.
- Figure 2a is improved to better explain the notation in our dynamical equations
- A table with p-values of the statistical test is added in the appendix to strengthen the interpretation of our results.
- A shorter description of the data generation process from the appendix is added to the main text to increase the readability.

Additionally, as suggested by reviewers fCck, we have added a new baseline (GloVe) to improve the quality of our paper. We utilized the code provided by ([https://nlp.stanford.edu/projects/glove/](https://nlp.stanford.edu/projects/glove/)) to implement GloVe on our problems. Since the method itself could not extract hierarchical information, we apply Hierarchical Chunking Phase on the cosine similarity matrix learned by GloVe. Different values of hyperparameters were tested to produce the best result. The table below showed the result of GloVe in imbalanced and dynamical hierarchical structure environments.

IH      | $0.428\pm0.046$

HB    | $0.383\pm0.128$

IEH   | $0.517\pm0.063$

DIH   | $0.280\pm0.089$

DCH   | $0.483\pm0.144$

EC2EH | $0.535\pm0.052$

EH2EC | $0.612\pm0.008$

DCS   | $0.326\pm0.108$


 We appended two figures as supplementary material to show the learned cosine similarity matrix and dendrogram produced from the matrix. In short, the result we obtained showed that GloVe falls short when compared to the Hierarchical TSFMap and even Word2vec when solving the same problems. The figures (cosine similarity matrix) demonstrated that the hierarchical information is not properly learned. The additional results and discussion of the new baseline will be shown in the revised paper.

Concerning the novelty of the paper, we would argue that the hierarchical problem is a challenging and unsolved problem. The complexity is furthermore increased by making the problem dynamic, which to the best of the authors’ knowledge, has never been solved before. We find this aspect to be overlooked by the reviewers. Besides, the implementation of our model on complex network data is merely a showcase of real-world applications. Despite being a common dataset, our method managed to uncover hierarchical information that is not widely discussed before. In fact, any serializable data, such as sequences of objects/events can be adopted as input and processed by our model to extract hierarchical information.

---

### Decision · Program_Chairs · 2023-01-20

**Decision:**

Reject

**Justification For Why Not Higher Score:**

This paper engages in some over-reach in its claims. It is important not to accept a paper that does this, as it could lead readers to not fully understand what the exact contribution is here.

**Justification For Why Not Lower Score:**

N/A

**Metareview: Summary, Strengths And Weaknesses:**

This paper presents an algorithm for hierarchical clustering from sequential data using a dynamic, self-organizing approach (e.g. no loss function is being minimized, at least not explicitly). The authors analyse the dynamics of the model in various conditions and also show that this algorithm can discover appropriate hierarchical structures in real-world network data.

The strength of this paper is that it provides a new method to tackle a problem that is by no means fully solved (hierarchical clustering from sequential data).

The principal weakness is that the paper makes fairly grand claims (e.g. that they outperform the SotA and that they are somehow shedding light on how self-organization can allow for intelligent behavior more broadly), when a careful reading of the paper shows that it is a fairly incremental improvement on previous work (namely Vargas & Asabuki, 2021, AAAI), which extends the previous approach to work with hierarchy.

This was a borderline case, with an average score of 5.25 (min 3, max 8). Based on the meeting with the reviewers (see below) it was agreed that though this paper makes some decent contributions, it is not sufficiently clear about these, and engages in some over-reach with respect to what it claims its contributions actually are. Most importantly, the authors should have been more transparent about the incremental nature of the work, avoid overly sweeping statements about the implications for neuroscience and AI more broadly, and including more data comparing to truly SotA systems. Though the authors attempted to address some of these issues (e.g. by including a comparison to BERT), the rebuttals did not change the scores greatly. Based on all of these considerations a decision of reject was reached. The authors are encouraged to consider these comments and resubmit to another venue, though.

**Summary Of Ac-Reviewer Meeting:**

The meeting focused on the question of whether the authors were being clear about the true contributions. Though one of the reviewers felt the paper was sufficiently interesting to consider it for acceptance, the others felt that it was insufficiently clear about what is actually being achieved here to accept it at ICLR. As an AC, I was convinced by the latter argument, and the positive reviewer felt that it was not wholly unfair. Thus, a reject decision was natural.